# An Extensible Framework for Open Heterogeneous Collaborative Perception

**Yifan Lu**[1,4], **Yue Hu**[1,4], **Yiqi Zhong**[2], **Dequan Wang**[1,3], **Yanfeng Wang**[1,3], **Siheng Chen**[1,3,4✉],

[1] Shanghai Jiao Tong University, [2] University of Southern California, [3] Shanghai AI Lab

[4] Multi-Agent Governance & Intelligence Crew (MAGIC)

[1] {yifan_lu, 18671129361, dequanwang, wangyanfeng, sihengc}@sjtu.edu.cn

[2] yiqizhon@usc.edu

## Abstract

Collaborative perception aims to mitigate the limitations of single-agent perception, such as occlusions, by facilitating data exchange among multiple agents. However, most current works consider a homogeneous scenario where all agents use identity sensors and perception models. In reality, heterogeneous agent types may continually emerge and inevitably face a domain gap when collaborating with existing agents. In this paper, we introduce a new open heterogeneous problem: *how to accommodate continually emerging new heterogeneous agent types into collaborative perception, while ensuring high perception performance and low integration cost?* To address this problem, we propose **HE**terogeneous **AL**liance (HEAL), a novel extensible collaborative perception framework. HEAL first establishes a unified feature space with initial agents via a novel multi-scale foreground-aware Pyramid Fusion network. When heterogeneous new agents emerge with previously unseen modalities or models, we align them to the established unified space with an innovative backward alignment. This step only involves individual training on the new agent type, thus presenting extremely low training costs and high extensibility. To enrich agents' data heterogeneity, we bring OPV2V-H, a new large-scale dataset with more diverse sensor types. Extensive experiments on OPV2V-H and DAIR-V2X datasets show that HEAL surpasses SOTA methods in performance while reducing the training parameters by 91.5% when integrating 3 new agent types. We further implement a comprehensive codebase at: https://github.com/yifanlu0227/HEAL.

## 1 Introduction

Multi-agent collaborative perception promotes better and more holistic perception by enabling multiple agents to share complementary perceptual information with each other (Wang et al., 2020; Xu et al., 2022c; Li et al., 2021). This task can fundamentally overcome several long-standing issues in single-agent perception, such as occlusion (Wang et al., 2020). The related methods and systems have tremendous potential in many applications, including multi-UAVs (unmanned aerial vehicles) for search and rescue (Hu et al., 2022), multi-robot automation and mapping (Carpin, 2008), and vehicle-to-vehicle (V2V) and vehicle-to-everything (V2X) collaboration.

In this emerging field, most current works (Lu et al., 2023; Lei et al., 2022) make a plausible, yet oversimplified assumption: all the agents have to be homogeneous; that is, all agents' perception systems use the same sensor modality and share the same detection model. However, in the real world, the modalities and models of agents are likely to be heterogeneous, and new agent types may continuously emerge. Due to the rapid iteration of sensor technologies and perception algorithms, coupled with the various attitudes of agent owners (like autonomous driving companies) towards collaborative perception, it is inherently challenging to definitively determine all agent types from the outset. When a heterogeneous agent, which has never appeared in the training set, wishes to join the collaboration, it inevitably encounters a domain gap with the existing agents. This gap substantially impedes its capability to fuse features with the existing collaborative agents and markedly limits the extensibility of the collaborative perception. Thus, the problem of **open heterogeneous collaborative perception** arises: *how to accommodate continually emerging new agent types into the existing collaborative perception while ensuring high perception performance and low integration cost?*

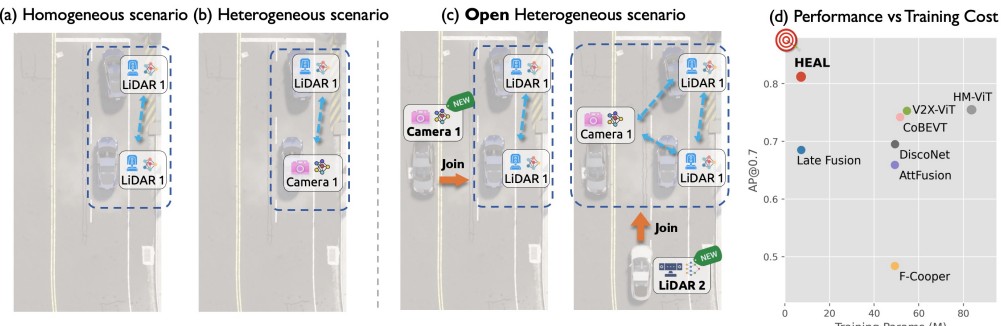

Figure 1: (**a**) homogeneous setting, where agents have identical modality and model. (**b**) heterogeneous setting, where agents' modalities and models are distinct but pre-determined. (**c**) Open heterogeneous setting, where new types of agents want to join collaboration with previously unseen modalities or models. (**d**) HEAL holds the SOTA performance while minimizing the training cost (model parameters here) when integrating a new agent type. The bullseye represents the best.

The designation *open heterogeneous* underscores the unpredictable essence of the incoming agent's modality and model; see Figure. 1 for an illustration. To address this issue, one viable solution is late fusion. By fusing each agent's detection outputs, late fusion bypasses the heterogeneity among new agents and existing agents. However, its performance is suboptimal and has been shown particularly vulnerable to localization noise (Lu et al., 2023) and communication latency (Wang et al., 2020). Another potential approach is fully collective training like HM-ViT (Xiang et al., 2023), which aggregates all agent types in training to fill domain gaps. However, this approach requires retraining the entire model every time a new agent type is introduced. This becomes increasingly training-expensive as new agents continuously emerge.

To address this open heterogeneous collaborative perception problem, we propose **HE**terogeneous **AL**liance (HEAL), a novel extensible framework that integrates new agent types into collaboration with ultra-low costs. The core idea is to sustain a unified feature space for multi-agent collaboration and ensure new agent types align their features to it. HEAL has two training phases: collaboration base training and new agent type training. In the first phase, HEAL sets initial agents as the collaboration base and undertakes collective end-to-end training to create a robust unified feature space for all agents. It uses the innovative *Pyramid Fusion*, a multi-scale and foreground-aware network, to fuse features and promote the learning of the unified space. In the next phase, when agents with a new heterogeneous type aim to join the collaboration, HEAL designs a novel *backward alignment* mechanism for their individual training. The inherited Pyramid Fusion module acts as new agents' detection back-end, with only new agents' front-end encoders updated. This prompts new agents to align their features with the unified feature space. Such individual training eliminates the high costs associated with collective retraining when adding new agent types, presenting extremely low model size, FLOPs, training time, and memory consumption. Further, it preserves the new agents' model and data privacy. As the backward alignment can be conducted locally, it protects new agents' model details and allows agent owners to use their sensor data for training, significantly addressing automotive companies' data- and model-privacy concerns. Once the training is complete, all agents of the new type can join the alliance with feature-level collaboration. By repeating the second phase, the alliance can continuously incorporate new types of agents as they emerge.

To evaluate HEAL and further promote open heterogeneous collaborative perception, we propose a large-scale heterogeneous collaborative perception dataset, OPV2V-H, which supplements more sensor types based on the existing OPV2V (Xu et al., 2022c). Extensive experiments on OPV2V-H and real-world dataset DAIR-V2X (Yu et al., 2022) show HEAL's remarkable performance. In the experiment of successively adding 3 types of heterogeneous agents, HEAL outperforms the other methods in collaborative detection performance while reducing 91.5% of the training parameters compared with SOTA. We summarize our contributions as follows:

- In considering the scenario of continually emerging new heterogeneous agents, we present HEAL, the first extensible heterogeneous collaborative perception framework. HEAL ensures extensibility by establishing a unified feature space and aligning new agents to it.

- We propose a powerful Pyramid Fusion for the collaboration base training, which utilized multiscale and foreground-aware designs to sustain a potent unified feature space.

- To integrate new types of agents, we introduce a novel backward alignment mechanism to align heterogeneous agents to the unified space. This training is conducted locally on single agents, reducing training costs while also preserving model details.

- We propose a new dataset OPV2V-H to facilitate the research of heterogeneous collaborative perception. Extensive experiments on OPV2V-H and real-world DAIR-V2X datasets demonstrate HEAL's SOTA performance and ultra-low training expenses.

## 2 RELATED WORKS

### 2.1 COLLABORATIVE PERCEPTION

The exchange of perception data among agents enables collaborative agents to achieve a more comprehensive perceptual outcome (Wang et al., 2020; Yu et al., 2022; Li et al., 2021; Hu et al., 2023; Wei et al., 2023; Liu et al., 2020a; Li et al., 2023). Early techniques transmitted either raw sensory data (known as early fusion) or perception outputs (known as late fusion). However, recent studies have been studying the transmission of intermediate features to balance performance and bandwidth. To boost the research of multi-agent collaborative perception, V2X-Sim (Li et al., 2022c) and OPV2V Xu et al. (2022c) generated high-quality simulation datasets, and DAIR-V2X (Yu et al., 2022) collects real-world data. To achieve an effective trade-off between perception performance and communication costs, Who2com (Liu et al., 2020b), When2com Liu et al. (2020a) and Where2comm (Hu et al., 2022) select the most critical message to communicate. To resist pose errors, Vadivelu et al. (2021) and Lu et al. (2023) use learnable or mathematical methods to correct the pose errors. Collaborative perception can also directly help the driving planning and control task Chen & Krähenbühl (2022); Cui et al. (2022); Zhu et al. (2023) with more accurate perceptual results. Most papers assume that agents are given the same sensor modality and model, which is deemed impractical in the real world. A contemporaneous work solving agent's modality heterogeneity is HM-ViT (Xiang et al., 2023), but it neglects the framework's extensibility and requires retraining the whole model when adding new agent types. In this paper, we address the issues of heterogeneity and extensibility together.

### 2.2 MULTI-MODALITY FUSION

In the field of 3D object detection, the fusion of LiDAR and camera data has demonstrated promising results (Chen et al., 2022a; Li et al., 2022d; Yang et al., 2022; Bai et al., 2022; Borse et al., 2023; Li et al., 2022b; Xu et al., 2022d): LiDAR-to-camera (Ma et al., 2019) methods project LiDAR points to camera planes. By using this technique, a sparse depth map can be generated and combined with image data. Camera-to-LiDAR (Vora et al., 2020; Li et al., 2022d) methods decorated LiDAR points with the color and texture information retrieved from images. Bird's eye view (BEV) (Liu et al., 2022a; Li et al., 2022a; Borse et al., 2023; Chen et al., 2022b) provides a spatial unified representation for different modalities to perform feature fusion. As stated in (Xiang et al., 2023), in single-agent multi-modality settings, sensor types/numbers and their relative poses are fixed, with most LiDAR-camera fusion algorithms (Chen et al., 2022a; Li et al., 2022d; Yang et al., 2022) developed based on this fact. In contrast, heterogeneous multi-agent collaboration has random sensor positions and types, differing fundamentally from single-vehicle multi-modality fusion. However, the aforementioned BEV representation is a highly efficacious approach to facilitate spatial alignment among agents of diverse modalities and models.

## 3 OPEN HETEROGENEOUS COLLABORATIVE PERCEPTION

Multi-agent collaborative perception allows a group of agents to collectively perceive the whole environment, exchanging complementary perceptual information with each other. Within this domain, *open heterogeneous collaborative perception* considers the scenario where new agent types with unseen sensor modalities or perception models can be continually added to the existing collaborative system. Without loss of generality, consider $N$ homogeneous agents in the scene initially. These agents, uniformly equipped with identical sensors and perception models, have the capability to observe, communicate, and compute, thereby fostering a homogeneous collaborative network. Subsequently, some new types of agents with previously unseen sensor modalities or models, are introduced into the scene sequentially to join the collaboration. Such dynamism characterizes the attributes of deploying collaborative perception in the real world: agent types will not be fully determined at the beginning and the number of types is likely to increase over time. It highly differs from traditional heterogeneous frameworks, where agent types are predetermined and fixed.

To tackle open heterogeneous collaborative perception, a straightforward approach is to conduct training for both existing and new agents collectively. However, this approach is computationally expensive with repetitive integrations over time. Therefore, an effective solution must balance two primary goals: i) minimizing the training overhead associated with each integration, and ii) maximizing perception performances across all agent types post-integration. The pursuit of these twin goals is fundamental to the successful implementation of open heterogeneous collaborative perception in diverse and dynamically changing real-world scenarios.

## 4 HETEROGENEOUS ALLIANCE (HEAL)

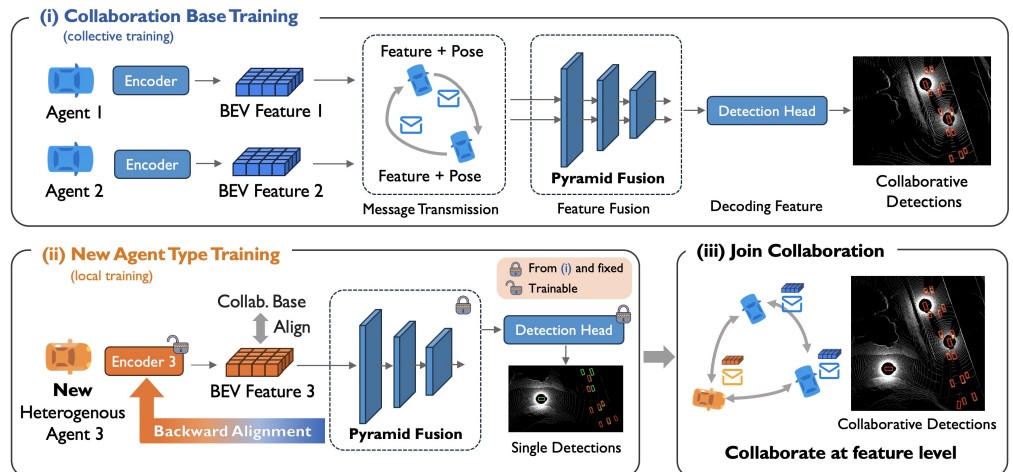

Figure 2: Overview of HEAL. (i) We train the initial homogeneous agents (collaboration base) with our novel Pyramid Fusion to establish a unified feature space; (ii) We leverage the well-trained Pyramid Fusion and detection head as the new agents' detection back-end. With the back-end fixed, it pushes the encoder to align its features within the unified feature. This step is performed on the new agent type only, presenting extremely low training costs. (iii) New agents join the collaboration.

To address open heterogeneous collaborative perception problem, we propose **HE**terogeneous **AL**liance (HEAL). It is an extensible framework to seamlessly integrate new agent types into the existing collaborative network with both minimal training overhead and optimal performance. As shown in Figure. 2, HEAL includes two phases to realize a growing alliance: i) collaboration base training, which allows initial agents to collaborate at feature-level and create a unified feature space; and ii) new agent type training, which aligns new agents' feature with the previously established unified feature space for collaboration. For every integration of a new agent type, only the second phase is required. We now elaborate on each training phase in the following subsections.

### 4.1 COLLABORATION BASE TRAINING

In this phase, we designate the initial homogeneous agents as our collaboration base and train a feature-level collaborative perception network. Let $\mathcal{S}_{[b]}$ be the set of $N$ agents with the same base agent type $b$. For the $i$th agent in the set $\mathcal{S}_{[b]}$, we denote $\mathbf{O}_i$ as it observation, $f_{\text{encoder}[b]}(\cdot)$ as its perception encoder and $\mathbf{B}_i$ as its final detection output. Then, the collaborative perception network of the $i$th agent works as follows:

$$
\begin{aligned}
\mathbf{F}_i &= f_{\text{encoder}[b]}\left(\mathbf{O}_i\right), & i \in \mathcal{S}_{[b]} & \quad \triangleright \text{Feature Encoding} & (1a)\\
\mathbf{F}_{j \to i} &= \Gamma_{j \to i}\left(\mathbf{F}_j\right), & j \in \mathcal{S}_{[b]} & \quad \triangleright \text{Message Transmission} & (1b)\\
\mathbf{H}_i &= f_{\text{pyramid\_fusion}}\left(\{\mathbf{F}_{j \to i}\}_{j \in \mathcal{S}_{[b]}}\right), & & \quad \triangleright \text{Feature Fusion} & (1c)\\
\mathbf{B}_i &= f_{\text{head}}(\mathbf{H}_i), & & \quad \triangleright \text{Decoding Feature} & (1d)
\end{aligned}
$$

where $\mathbf{F}_i$ is the initial feature map from the encoder with BEV representation, $\Gamma_{j \to i}(\cdot)$ is an operator that transmits $j$th agent's feature to the $i$th agent and performs spatial transformation, $\mathbf{F}_{j \to i}$ is the spatially aligned BEV feature in $i$th's coordinate (note that $\mathbf{F}_{i \to i} = \mathbf{F}_i$), $\mathbf{H}_i$ is the fused feature and $\mathbf{B}_i$ is the final detection output obtained by a detection head $f_{\text{head}}(\cdot)$.

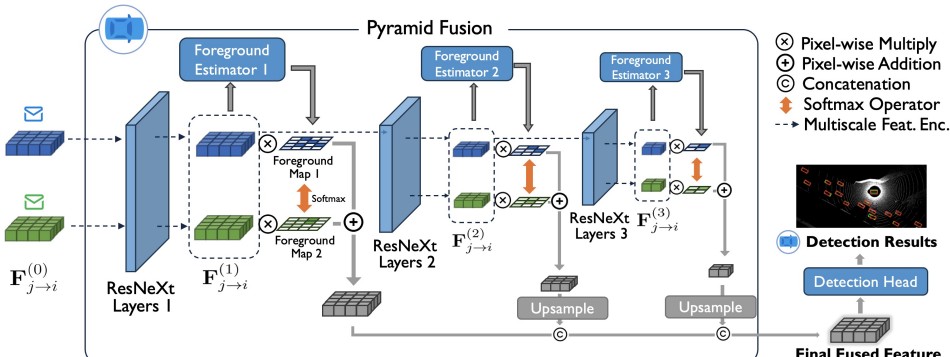

Figure 3: Pyramid Fusion uses multiscale and foreground-aware designs to fuse features and create a robust unified feature space. Foreground estimators produce foreground possibility maps at each BEV position. These foreground maps are then normalized to weights for feature summation. Foreground maps are subject to supervision during training. Blue and green represent different agents.

To provide a well-establish feature space for multi-agent collaboration, we propose a novel Pyramid Fusion $f_{\text{pyramid\_fusion}}(\cdot)$ in Step ( 1c), designed with multiscale and foreground-aware structure. Let $\mathbf{F}_{j \to i}^{(0)}$ be $\mathbf{F}_{j \to i}$, we elaborate $f_{\text{pyramid\_fusion}}(\cdot)$ in detail:

$$
\begin{aligned}
\mathbf{F}_{j \to i}^{(\ell)} &= R_\ell(\mathbf{F}_{j \to i}^{(\ell-1)}), & j \in \mathcal{S}_{[b]}, \text{ and } \ell = 1, 2, \cdots, L, & \quad (2a) \\
\mathbf{S}_{j \to i}^{(\ell)} &= H_\ell(\mathbf{F}_{j \to i}^{(\ell)}), & j \in \mathcal{S}_{[b]}, \text{ and } \ell = 1, 2, \cdots, L, & \quad (2b) \\
\mathbf{W}_{j \to i}^{(\ell)} &= \texttt{softmax}(\mathbf{S}_{j \to i}^{(\ell)}), & j \in \mathcal{S}_{[b]}, \text{ and } \ell = 1, 2, \cdots, L, & \quad (2c) \\
\mathbf{F}_{i}^{(\ell)} &= \texttt{sum}(\{\mathbf{F}_{j \to i}^{(\ell)} * \mathbf{W}_{j \to i}^{(\ell)}\}_{j \in \mathcal{S}_{[b]}}), & & \quad (2d) \\
\mathbf{H}_{i} &= \texttt{concat}([\mathbf{F}_{i}^{(1)}, u_2(\mathbf{F}_{i}^{(2)}), \cdots, u_L(\mathbf{F}_{i}^{(L)})]), & & \quad (2e)
\end{aligned}
$$

where $\ell$ indicates the scale, $R_\ell(\cdot)$ is the $\ell$th ResNeXt (Xie et al., 2017) layer with a downsampling rate of 2, $\mathbf{F}_{j \to i}^{(\ell)}$ is encoded features at $\ell$th scale; $H_\ell(\cdot)$ is $\ell$th foreground estimator that outputs the foreground map $\mathbf{S}_{j \to i}^{(\ell)}$, measuring the possibility of foreground object at each BEV position; $\texttt{softmax}(\cdot)$ normalizes the foreground possibility to the weights for multi-agent feature fusion $\mathbf{W}_{j \to i}^{(\ell)}$; $u_\ell(\cdot)$ is an upsampling operator for the $\ell$th scale.

The proposed Pyramid Fusion facilitates the establishment of a unified feature space for multi-agent collaboration by harnessing two key designs: a multi-scale structure and foreground awareness. First, the multi-scale ResNeXt layers create a comprehensive unified feature space by fusing features at varying BEV scales. This not only promotes feature fusion in the collaboration base, but also ensures adaptability for future new agents, allowing their alignment to this unified feature space with coarse and fine grain. Furthermore, fusing at higher scales mitigates the discretization errors introduced by spatial transformation, thereby enhancing the robustness of multi-agent feature fusion and future alignment. Second, the foreground awareness design leverages foreground estimators to obtain foreground maps, which will guide Pyramid Fusion to select the perceptually critical features for fusion. It also enables the the model to learn how to differentiate between the foreground and the background, leading to a more robust unified space.

To train the collaborative perception model for the collaboration base, the overall loss is:

$$
L = L_{\text{det}}(\mathbf{B}_i, \mathbf{Y}_i) + \sum_{\ell=1}^{L} \sum_{j=1}^{\mathcal{S}_{[b]}} \alpha_\ell L_{\text{focal}}\left(\mathbf{S}_{j \to i}^{(\ell)}, \mathbf{Y}_{j \to i}^{(\ell)}\right). \tag{3}
$$

The first term is detection supervision, where $L_{\text{det}}(\cdot)$ is the detection loss, including focal loss (Lin et al., 2017) for classification and Smooth-$L_1$ loss (Girshick, 2015) for regression, $\mathbf{Y}_i$ is the ground-truth detections and $\mathbf{B}_i$ is the detection output by our model. In addition to the supervision on the collaborative detection, we design the foreground map supervision at multiple BEV scales, where $L_{\text{focal}}$ refers to focal loss (Lin et al., 2017), $\mathbf{S}_{j \to i}^{(\ell)}$ is the estimated foreground map from Step ( 2b)

and $\mathbf{Y}_{j \to i}^{(\ell)}$ is the ground-truth BEV mask of foreground object for each agent. The hyperparameter $\alpha_\ell$ controls the effect of foreground supervision at various scales.

## 4.2 New Agent Type Training

Now we consider the integration of a new heterogeneous agent type, leveraging a novel backward alignment. The core idea is to utilize the Pyramid Fusion module and detection head from the previous phase as new agents' single detection back-end and only update the front-end encoder module, prompting the encoder to generate features with the pre-established unified feature space.

Specifically, we denote the new agent type as $n_1$ and new agent set as $\mathcal{S}_{[n_1]}$, and the full set of current agents set becomes $\mathcal{S} = \mathcal{S}_{[b]} \cup \mathcal{S}_{[n_1]}$. For agent $k$ in the agent set $\mathcal{S}_{[n_1]}$, we define $\mathbf{O}_k$ as its observation and $f_{\text{encoder}[n_1]}(\cdot)$ as its detector encoder. We keep $f_{\text{pyramid\_fusion}}^*(\cdot)$ and $f_{\text{head}}^*(\cdot)$ from previous stage unchanged, where $*$ denotes fixed, and train $f_{\text{encoder}[n_1]}(\cdot)$ **on single agents**:

$$\mathbf{F}_k = f_{\text{encoder}[n_1]}(\mathbf{O}_k), \qquad k \in \mathcal{S}_{[n_1]}, \tag{4a}$$

$$\mathbf{F}_k' = f_{\text{pyramid\_fusion}}^*(\mathbf{F}_k), \qquad k \in \mathcal{S}_{[n_1]}, \tag{4b}$$

$$\mathbf{B}_k = f_{\text{head}}^*(\mathbf{F}_k'), \qquad k \in \mathcal{S}_{[n_1]}, \tag{4c}$$

where $\mathbf{F}_k$ is the feature encoded from the new sensor and model, $\mathbf{F}_k'$ is the feature encoded by Pyramid Fusion module and $\mathbf{B}_k$ is the corresponding detections. Note that here we perform individual training for single agents; thus the input to the Pyramid Fusion is single-agent feature map $\mathbf{F}_k$ in (4b) instead of multi-agent feature maps $\{\mathbf{F}_{j \to i}\}_{j \in \mathcal{S}_{[b]}}$ as in (1c). With the pretrained Pyramid Fusion module and the detection head established as the back-end and **fixed**, the training process naturally evolves into adapting the front-end encoder $f_{\text{encoder}[n_1]}(\cdot)$ to the back-end's parameters, thereby enabling new agent types to align with unified space.

Our backward alignment works well for two reasons. First, BEV representation offers a shared co-ordinate system for varied sensors and models. Second, with the intrinsic design of Pyramid Fusion, feature domain alignment can be conducted with high efficiency: i) The alignment is performed across multiple scales, capturing and bridging the plausible feature scale differences between different modalities and models. ii) The foreground estimators are also retained, thereby preserving effective supervision on the alignment with the most important foreground feature.

In addition to enabling new and existing agents to collaborate at the feature level with robust performance, our backward alignment also shows a unique advantage: Training is only conducted on single agents of the new type. This significantly reduces the training costs of each integration, avoiding collecting spatio-temporal synchronized sensor data for multi-agent collective training and expensive retraining. Further, it prevents the new agents' model details from disclosure and allows the owner of the new agents can use their own sensor data. It would remarkably address many privacy concerns automotive companies might have when deploying the V2V techniques.

To supervise the training of new agent type with backward alignment, the loss is the same as Eq. ( 3), but we refer the detection bounding boxes $\mathbf{B}_i$ and ground-truth bounding boxes $\mathbf{Y}_i$ to belong to the single agents. The supervision on the confidence score at different scales is also preserved.

## 4.3 HEAL during Inference

Once the new agent of agent type $n_1$ has trained its encoder, all agents of new agent type $n_1$ can collaborate with base agents in the scene. Mathematically, for Agent $i$ in the set $\mathcal{S}_{[b]} \cup \mathcal{S}_{[n_1]}$, its feature after multi-agent fusion is obtained as $\mathbf{H}_i = f_{\text{pyramid\_fusion}}^*\left(\{\Gamma_{j \to i}(\mathbf{F}_j)\}_{j \in \mathcal{S}_{[b]} \cup \mathcal{S}_{[n_1]}}\right)$. This is feasible because two training phases ensure that for all $i, j$, $\mathbf{F}_j$ lies in the same feature space.

Following this, we can continually integrate emerging heterogeneous agents into our alliance by revisiting the steps outlined in Sec. 4.2, creating a highly extensible and expansive heterogeneous alliance. Assuming there are a total of $T$ new agent types, once training for each of these new agent types is completed, the collaboration among all heterogeneous agents can be written as:

$$\mathbf{H}_i = f_{\text{pyramid\_fusion}}^*\left(\{\Gamma_{j \to i}(\mathbf{F}_j)\}_{j \in \mathcal{S}_{[b]} \cup \mathcal{S}_{[n_1]} \cup \mathcal{S}_{[n_2]} \cup \cdots \cup \mathcal{S}_{[n_T]}}\right).$$

Then, we can decode the feature and obtain the final detections $\mathbf{B}_i = f_{\text{head}}^*(\mathbf{H}_i)$.

## 5 EXPERIMENTAL RESULTS

### 5.1 DATASETS

**OPV2V-H.** We propose a simulation dataset dubbed OPV2V-H. In OPV2V (Xu et al., 2022c) dataset, the LiDAR with 64-channel shows a significant detection advantage over the camera modality. Evaluations of heterogeneous collaboration on OPV2V may not truly represent the collaboration performance between these two modalities since LiDAR agents can provide most detections (Xiang et al., 2023). For this purpose, we collected more data to bridge the gap between LiDAR and camera modalities, leading to the new OPV2V-H dataset. OPV2V-H dataset has on average approximately 3 agents with a minimum of 2 and a maximum of 7 in each frame. Except for one 64-channel LiDAR and four 4 RGB cameras (resolution 800*600) of each agent from original OPV2V dataset, OPV2V-H collects extra 16- and 32-channel LiDAR data and 4 depth camera data.

**DAIR-V2X.** DAIR-V2X (Yu et al., 2022) is a real-world collaborative perception dataset. The dataset has 9K frames featuring one vehicle and one roadside unit (RSU), both equipped with a LiDAR and a 1920x1080 camera. RSU' LiDAR is 300-channel while the vehicle's is 40-channel.

### 5.2 OPEN HETEROGENEOUS SETTINGS

We consider the sequential integration of new heterogeneous agents into the collaborative system to evaluate the collaborative performance and training costs of HEAL. We prepared four agent types, including 2 LiDAR models and 2 camera models; see Table. 1:

Table 1: Agent type setting for open heterogeneous collaborative perception experiments.

| Agent Type | Agent **Sensor** and *Model* Setup |
|---|---|
| $\mathbf{L}_P^{(x)}$ | **LiDAR** of $x$-channel, *PointPillars* (Lang et al., 2019). |
| $\mathbf{C}_E^{(x)}$ | **Camera**, resize img. to height $x$ px , Lift-Splat (Philion & Fidler, 2020) w. *EfficientNet* (Tan & Le, 2019) as img. encoder. |
| $\mathbf{L}_S^{(x)}$ | **LiDAR** of $x$-channel, *SECOND* (Yan et al., 2018). |
| $\mathbf{C}_R^{(x)}$ | **Camera**, resize img. to height $x$ px , Lift-Splat (Philion & Fidler, 2020) w. *ResNet50* (He et al., 2016) as img. encoder. |

**Implementation details.** We first adopt $\mathbf{L}_P^{(64)}$ agents as the collaboration base to establish the unified feature space. And then select $(\mathbf{C}_E^{(384)}, \mathbf{L}_S^{(32)}, \mathbf{C}_R^{(336)})$ to be new agent types. Both PointPillars (Lang et al., 2019), SECOND (Yan et al., 2018) and Lift-Splat-Shoot (Philion & Fidler, 2020) encode input data with grid size $[0.4m, 0.4m]$, and further downsample the feature map by $2\times$ and shrink the feature dimension to 64 with 3 ConvNeXt (Liu et al., 2022b) blocks for message sharing. The multi-scale feature dimension of Pyramid Fusion is [64,128,256]. The ResNeXt layers have [3,5,8] blocks each. Foreground estimators are $1 \times 1$ convolution with channel [64,128,256]. The hyper-parameter $\alpha_\ell = \{0.4, 0.2, 0.1\}_{\ell=1,2,3}$. We incorporated depth supervision for all camera detections to help convergence. We train the collaboration base and new agent types both for 25 epochs end-to-end with Adam, reducing the learning rate from 0.002 by 0.1 at epochs 15. Training costs 5 hours for the collaboration base on 2 RTX 3090 GPUs and 3 hours for each new agent's training, while Xiang et al. (2023) takes more than 1 day to converge with 4 agent types together. We adopt Average precision (AP) at different Intersection-over-Union (IoU) to measure the perception performance. The training range is $x \in [-102.4m, +102.4m]$, $y \in [-51.2m, +51.2m]$ but we expand the range to $x \in [-204.8m, +204.8m]$, $y \in [-102.4m, +102.4m]$ in evaluation for holistic view. Late Fusion aggregates all detected boxes from single agents. Considering most scenarios in the test set involve fewer than 4 agents, we initialized the scenario with one $\mathbf{L}_P^{(64)}$ agent and progressively introduced $\mathbf{C}_E^{(384)}, \mathbf{L}_S^{(32)}, \mathbf{C}_R^{(336)}$ agents into the scene for evaluation.

### 5.3 QUANTITATIVE RESULTS

**Performance and training cost.** Table. 2 compares the detection performance and training cost. We see that HEAL surpassed all existing collaborative perception methods in perception performances while maintaining the lowest training cost. This is attributed to our powerful Pyramid Fusion structure and cost-efficient backward alignment design. Pyramid Fusion selects the most important feature for fusion in a multiscale manner and sustains a potent unified space. Backward alignment further reduces the training costs of integrating new agents remarkably. This step does not involve collective training, thus ensuring consistent and extremely low training costs. It presents significant advantages in model size, FLOPs, training time, and memory usage. In contrast, other baseline methods necessitate retraining all models at each integration with a computational complexity of

Table 2: We evaluate the performance and training cost on OPV2V-H dataset when increasingly adding 3 new heterogeneous agents to the scene for collaboration (starting with one $\mathbf{L}_P^{(64)}$ agent). Every method requires retraining the whole model except for 'no fusion', 'late fusion', and HEAL. Metrics related to the training cost are all measured with batchsize 1 on 1 RTX A40. Optimal values among intermediate fusion methods are bolded. Percentage comparison is made with HM-ViT.

| Metric | AP50 ↑ | | | AP70 ↑ | | | Train Throughput (#/sec.) ↑ | | |
|---|---|---|---|---|---|---|---|---|---|
| Based on $\mathbf{L}_P^{(64)}$, Add New Agent | $+\mathbf{C}_E^{(384)}$ | $+\mathbf{L}_S^{(32)}$ | $+\mathbf{C}_R^{(336)}$ | $+\mathbf{C}_E^{(384)}$ | $+\mathbf{L}_S^{(32)}$ | $+\mathbf{C}_R^{(336)}$ | $+\mathbf{C}_E^{(384)}$ | $+\mathbf{L}_S^{(32)}$ | $+\mathbf{C}_R^{(336)}$ |
| No Fusion | 0.748 | 0.748 | 0.748 | 0.606 | 0.606 | 0.606 | / | / | / |
| Late Fusion | 0.775 | 0.833 | 0.834 | 0.599 | 0.685 | 0.685 | 3.29 | 5.75 | 4.75 |
| F-Cooper (Chen et al., 2019) | 0.778 | 0.742 | 0.761 | 0.628 | 0.517 | 0.494 | 2.41 | 2.93 | 2.54 |
| DiscoNet (Li et al., 2021) | 0.798 | 0.833 | 0.830 | 0.653 | 0.682 | 0.695 | 2.37 | 2.87 | 2.47 |
| AttFusion (Xu et al., 2022c) | 0.796 | 0.821 | 0.813 | 0.635 | 0.685 | 0.659 | 2.36 | 2.90 | 2.39 |
| V2XViT (Xu et al., 2022b) | 0.822 | 0.888 | 0.882 | 0.655 | 0.765 | 0.753 | 1.45 | 1.58 | 1.45 |
| CoBEVT (Xu et al., 2022a) | 0.822 | 0.885 | 0.885 | 0.671 | 0.742 | 0.742 | 1.94 | 2.27 | 1.99 |
| HM-ViT (Xiang et al., 2023) | 0.813 | 0.871 | 0.876 | 0.646 | 0.743 | 0.755 | 1.22 | 1.33 | 1.18 |
| HEAL | **0.826** | **0.892** | **0.894** | **0.726** | **0.812** | **0.813** | **3.27** | **5.44** | **4.59** (↑ 2.88×) |

| Metric | Model #Params (M) ↓ | | | FLOPs(T) ↓ | | | Peak Memory (GB) ↓ | | |
|---|---|---|---|---|---|---|---|---|---|
| Based on $\mathbf{L}_P^{(64)}$, Add New Agent | $+\mathbf{C}_E^{(384)}$ | $+\mathbf{L}_S^{(32)}$ | $+\mathbf{C}_R^{(336)}$ | $+\mathbf{C}_E^{(384)}$ | $+\mathbf{L}_S^{(32)}$ | $+\mathbf{C}_R^{(336)}$ | $+\mathbf{C}_E^{(384)}$ | $+\mathbf{L}_S^{(32)}$ | $+\mathbf{C}_R^{(336)}$ |
| No Fusion | / | / | / | / | / | / | / | / | / |
| Late Fusion | 20.25 | 6.34 | 7.15 | 0.149 | 0.074 | 0.091 | 3.70 | 2.11 | 2.72 |
| F-Cooper (Chen et al., 2019) | 30.70 | 39.59 | 49.22 | 0.190 | 0.263 | 0.322 | 20.27 | 14.56 | 18.36 |
| DiscoNet (Li et al., 2021) | 30.80 | 39.67 | 49.29 | 0.194 | 0.270 | 0.332 | 19.92 | 17.10 | 15.95 |
| AttFusion (Xu et al., 2022c) | 30.78 | 39.60 | 49.22 | 0.190 | 0.263 | 0.323 | 20.18 | 13.88 | 19.77 |
| V2XViT (Xu et al., 2022b) | 36.17 | 44.99 | 54.62 | 0.315 | 0.352 | 0.393 | 25.92 | 21.25 | 21.00 |
| CoBEVT (Xu et al., 2022a) | 33.24 | 42.05 | 51.68 | 0.243 | 0.280 | 0.321 | 20.94 | 20.16 | 23.29 |
| HM-ViT (Xiang et al., 2023) | 47.71 | 65.08 | 83.34 | 0.236 | 0.331 | 0.442 | 35.53 | 26.55 | 26.88 |
| HEAL | **20.25** | **6.34** | **7.15** (↓ 91.5%) | **0.149** | **0.074** | **0.091** (↓ 79.5%) | **3.71** | **2.15** | **2.76** (↓ 89.8%) |

Table 3: Heterogeneous type agents are added in the order presented from left to right in each type combination. $\sum$ M.#P. represents the accumulation of model parameters for collaboration base training and each integration. Percentage comparison is made with HM-ViT. HEAL holds the best performance and the lowest training cost under various agent type combinations.

| Dataset | OPV2V-H (4 agents) | | | | DAIR-V2X (2 agents) | | | | | |
|---|---|---|---|---|---|---|---|---|---|---|
| Agent Types | $\mathbf{L}_P^{(32)}+\mathbf{C}_E^{(384)}+\mathbf{L}_S^{(32)}+\mathbf{C}_R^{(336)}$ | | $\mathbf{L}_P^{(16)}+\mathbf{C}_E^{(384)}+\mathbf{L}_S^{(16)}+\mathbf{C}_R^{(336)}$ | | $\mathbf{L}_P^{(40)}+\mathbf{C}_E^{(288)}$ | | $\mathbf{L}_P^{(40)}+\mathbf{L}_S^{(40)}$ | | $\mathbf{L}_P^{(40)}+\mathbf{C}_R^{(288)}$ | |
| Metric | AP70 ↑ | $\sum$ M.#P. ↓ | AP70 ↑ | $\sum$ M.#P. ↓ | AP50 ↑ | $\sum$ M.#P. ↓ | AP50 ↑ | $\sum$ M.#P. ↓ | AP50 ↑ | $\sum$ M.#P. ↓ |
| No Fusion | 0.504 | 5.47 | 0.281 | 5.47 | 0.392 | 5.47 | 0.392 | 5.47 | 0.392 | 5.47 |
| Late Fusion | 0.639 | 39.2 | 0.457 | 39.2 | 0.344 | 25.7 | 0.376 | 11.8 | 0.341 | 12.6 |
| F-Cooper | 0.430 | 127.6 | 0.299 | 127.6 | 0.626 | 38.8 | 0.545 | 24.9 | 0.611 | 25.8 |
| DiscoNet | 0.612 | 127.9 | 0.420 | 127.9 | 0.576 | 38.9 | 0.634 | 25.1 | 0.621 | 25.9 |
| AttFusion | 0.571 | 127.7 | 0.394 | 127.7 | 0.649 | 38.8 | 0.661 | 24.7 | 0.623 | 25.8 |
| V2XViT | 0.688 | 149.2 | 0.498 | 149.2 | 0.654 | 49.6 | 0.709 | 35.7 | 0.669 | 36.6 |
| CoBEVT | 0.682 | 137.5 | 0.491 | 137.5 | 0.638 | 43.8 | 0.692 | 29.8 | 0.660 | 30.7 |
| HM-ViT | 0.696 | 216.3 | 0.506 | 216.3 | 0.638 | 67.8 | 0.538 | 53.9 | 0.677 | 54.8 |
| HEAL | **0.738** | **39.2** (↓ 81.9%) | **0.578** | **39.2** (↓ 81.9%) | **0.658** | **25.7** (↓ 62.1%) | **0.770** | **11.8** (↓ 78.1%) | **0.681** | **12.6** (↓ 77.1%) |

O($m$) or even O($m^2$) (Xiang et al., 2023), where $m$ denotes the number of agent types. This limits their scalability, especially when there are numerous and growing types. Here when adding $\mathbf{C}_R^{(336)}$ agent, HEAL outperforms previous SOTA HM-ViT by 7.6% in AP70 with only 8.3% parameters.

**Agent type combination.** We present the final collaboration performance and accumulated training parameters with different agent type combinations on OPV2V-H and DAIR-V2X datasets in Table. 3. Specifically, we configure the LiDAR agent with lower channel numbers to show the performance on degraded LiDAR scenarios. Experiments show that no matter what kind of agent combination, HEAL always maintains the best performance and the lowest training cost.

**Imperfect localization.** Existing experiments hold the assumption that each agent has an accurate pose. However, in real-world scenarios, due to the presence of localization noise, features between agents might not align precisely. Consequently, we introduce a robustness experiment against pose errors, adding Gaussian noise to the accurate pose, as depicted in Figure. 4. Experimental results show that HEAL retains state-of-the-art performance even under various pose error conditions.

**Feature compression.** We use an autoencoder to reduce feature channels for bandwidth saving. Using well-trained HEAL and baseline methods, we finetune the new autoencoder. The compression ratio indicates the reduction in channels. Results in Figure. 4 demonstrate that even with a 32-fold compression, we still retain exceptionally high performance, surpassing baseline methods.

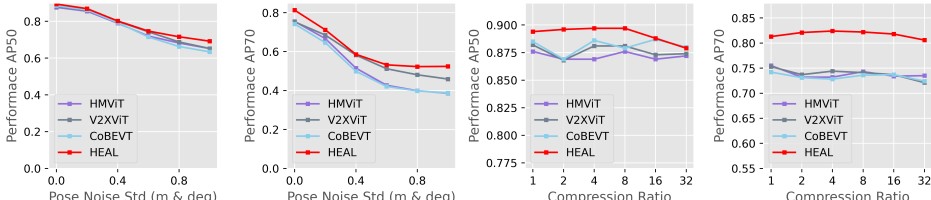

Figure 4: Robust Experiment to pose error and compression ratio. Pose noise is set to $\mathcal{N}(0, \sigma_p^2)$ on x, y location and $\mathcal{N}(0, \sigma_r^2)$ on yaw angle.

**Component ablation.** We carried out ablation experiments on HEAL's components and design, as shown in Table. 4. Results show that all of them are highly beneficial to collaboration performance. Through multiscale pyramid feature encoding at various scales, HEAL is capable of learning features at different resolutions and subsequently aligning new agent types via backward alignment. Further, the foreground supervision can help the HEAL distinguish the foreground from the background and select the most important features. These components help to construct a robust unified feature space and realize the alignment comprehensively.

Table 4: Ablation study of multiscale pyramid, foreground supervision, and backward alignment.

| Agent Types: $\mathbf{L}_P^{(64)} + \mathbf{C}_E^{(384)} + \mathbf{L}_S^{(32)} + \mathbf{C}_R^{(336)}$ | | | | |
|:---:|:---:|:---:|:---:|:---:|
| **Multiscale Pyramid** | **Foreground Supervision** | **Back. Align.** | AP50 ↑ | AP70 ↑ |
| | | | 0.729 | 0.578 |
| ✓ | | | 0.746 | 0.628 |
| | ✓ | | 0.753 | 0.633 |
| ✓ | ✓ | | 0.764 | 0.662 |
| | | ✓ | 0.841 | 0.732 |
| ✓ | | ✓ | 0.864 | 0.781 |
| | ✓ | ✓ | 0.879 | 0.782 |
| ✓ | ✓ | ✓ | **0.894** | **0.813** |

## 5.4 QUALITATIVE VISUALIZATIONS

HEAL makes features align within the same domain via backward alignment in Figure. 5 and achieves the best detection results in Figure. 6. L1,C1,L2 refer to $\mathbf{L}_P^{(64)}$, $\mathbf{C}_E^{(384)}$, $\mathbf{L}_S^{(32)}$ respectively.

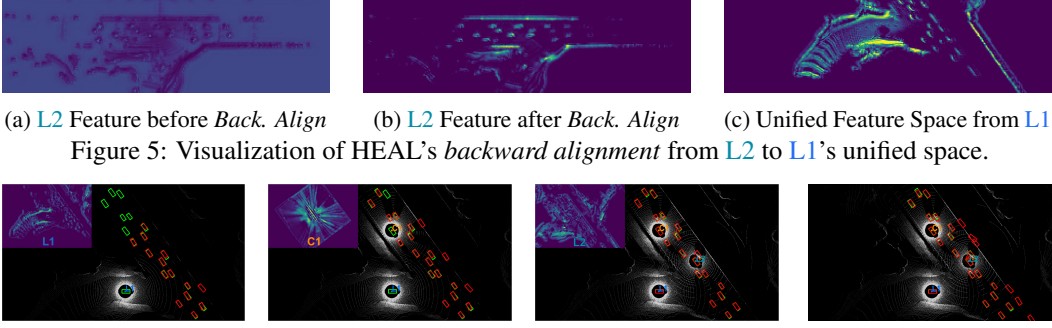

(a) L2 Feature before *Back. Align*    (b) L2 Feature after *Back. Align*    (c) Unified Feature Space from L1

Figure 5: Visualization of HEAL's *backward alignment* from L2 to L1's unified space.

(a) HEAL (L1)    (b) HEAL (L1+C1)    (c) HEAL (L1+C1+L2)    (d) HMViT (L1+C1+L2)

Figure 6: Visualization of open heterogeneous collaborative perception results on OPV2V-H. In (a)(b)(c), we show the process of gradually adding new agents to HEAL. The features of the added agent are displayed in the upper left corner. Note that the point cloud of the camera agent C1 is only used to indicate the agent's position. We color the predicted and GT boxes.

## 6 CONCLUSION

This paper proposes HEAL, a novel framework for open heterogeneous collaborative perception. HEAL boasts exceptional collaborative performance, minimal training costs and model-detail protection. Experiments on our proposed OPV2V-H and DAIR-V2X datasets validate the efficiency of HEAL, offering a practical solution for extensible collaborative perception deployment in the real-world scenario. The limitation of HEAL is that it requires BEV features, which requires extra effort for some models (e.g., keypoint-based LiDAR detection) to be compatible to the framework.

## 7 ACKNOWLEDGE

This research is supported by the National Key R&D Program of China under Grant 2021ZD0112801, NSFC under Grant 62171276 and the Science and Technology Commission of Shanghai Municipal under Grant 21511100900, 22511106101 and 22DZ2229005.

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

# A APPENDIX

## A.1 REAL-WORLD SOLUTION DETAILS

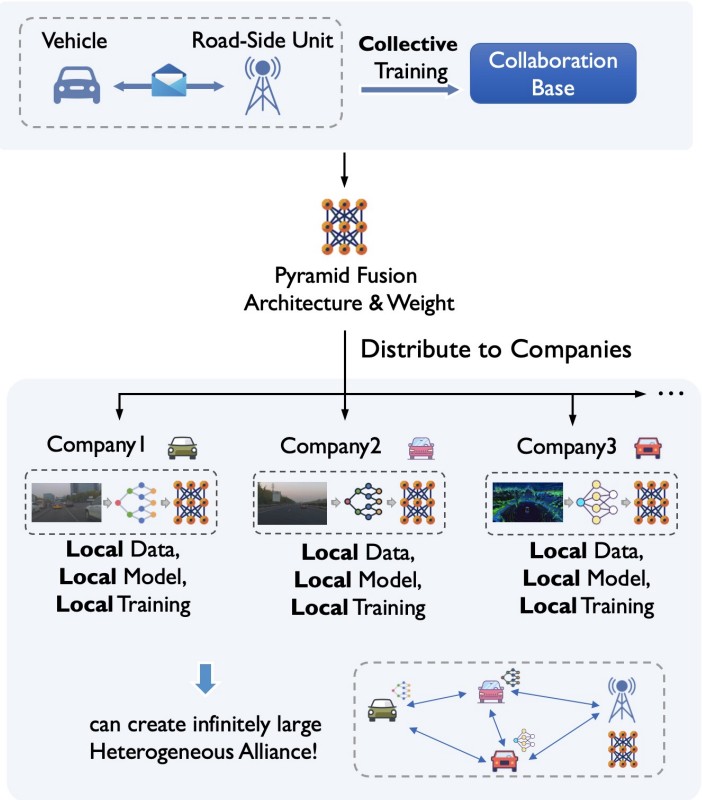

Figure 7: Real-world solution illustration. Take the DAIR-V2X dataset as an example, which consists of one vehicle and one Road-side Unit(RSU). We first trained the Pyramid Fusion using the vehicle and the RSU's data as the collaboration base. Subsequently, we distributed the vehicle's raw data and the Pyramid Fusion's weights to various companies, allowing them to train their respective models locally.

Deploying collaborative sensing in the real world is not a trivial task, largely due to the challenges in collecting training data. It necessitates the acquisition of multi-vehicle sensor data that has temporal and spatial synchronization, which is often a formidable endeavor. Precisely for these reasons, existing real-world collaborative perception datasets (Yu et al., 2022; Xu et al., 2023) typically encompass no more than 2 agents.

Given 2 agents in the scene, existing intermediate fusion methods is difficult to train three or more heterogeneous agent types together. However, due to our extensible design, HEAL shows the potential to achieve collaboration between infinite agent types even with the only 2 agent data. We think this is a very efficient training paradigm in real-world practice for automotive companies to join V2X collaborative perception, and it has great potential in the future. See Figure. 7 for illustration.

Take the DAIR-V2X dataset as an example, We can collectively train the Pyramid Fusion using vehicle-side and RSU-side LiDAR data. To expand the alliance, we distributed the Pyramid Fusion (and detection head)'s weights to companies that want to join the V2X collaboration. They can integrate their own vehicle's model encoder with the Pyramid Fusion module and train their model locally use their own data. After the backward alignment, they can collaborate with RSU at feature-level for robust and holistic perception. It is noteworthy that the **data, model, and associated training are all conducted locally**. This approach significantly safeguards the technical privacy of autonomous driving companies while simultaneously reducing training costs.

## A.2 VISUALIZATION OF FEATURE ALIGNMENT

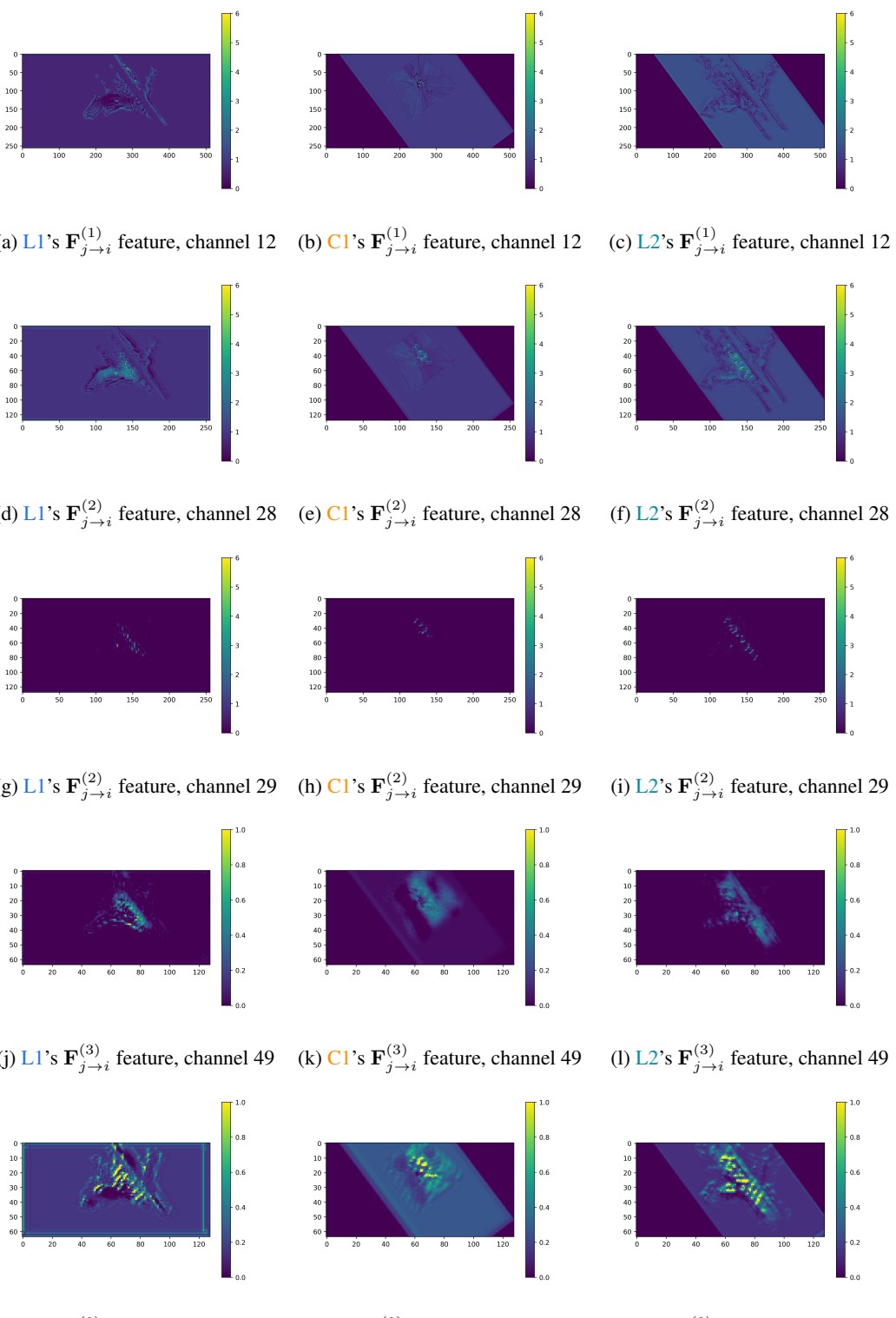

(a) L1's $\mathbf{F}_{j\to i}^{(1)}$ feature, channel 12    (b) C1's $\mathbf{F}_{j\to i}^{(1)}$ feature, channel 12    (c) L2's $\mathbf{F}_{j\to i}^{(1)}$ feature, channel 12

(d) L1's $\mathbf{F}_{j\to i}^{(2)}$ feature, channel 28    (e) C1's $\mathbf{F}_{j\to i}^{(2)}$ feature, channel 28    (f) L2's $\mathbf{F}_{j\to i}^{(2)}$ feature, channel 28

(g) L1's $\mathbf{F}_{j\to i}^{(2)}$ feature, channel 29    (h) C1's $\mathbf{F}_{j\to i}^{(2)}$ feature, channel 29    (i) L2's $\mathbf{F}_{j\to i}^{(2)}$ feature, channel 29

(j) L1's $\mathbf{F}_{j\to i}^{(3)}$ feature, channel 49    (k) C1's $\mathbf{F}_{j\to i}^{(3)}$ feature, channel 49    (l) L2's $\mathbf{F}_{j\to i}^{(3)}$ feature, channel 49

(m) L1's $\mathbf{F}_{j\to i}^{(3)}$ feature, channel 123  (n) C1's $\mathbf{F}_{j\to i}^{(3)}$ feature, channel 123    (o) L2's $\mathbf{F}_{j\to i}^{(3)}$ feature, channel 123

Figure 8: Visualization of the feature after ResNeXt Layer 1, 2 and 3 within the Pyramid Fusion. Visualization shows that they are in the same feature space and have similar response values for foreground and background pixels, which allows them to fuse together.

## A.3   PERFORMANCE OF PYRAMID FUSION ON HOMOGENEOUS SETTING

We provide the collaborative perception performance comparison with existing methods under the homogeneous setting, where all agents are equipped with the same sensor and detection model. Such benchmark has appeared in a number of existing works (Xu et al., 2022c; Xiang et al., 2023), which offers an intuitive performance comparison of fusion modules. We conduct experiments on LiDAR-based homogeneous collaboration and camera-based homogeneous collaboration to show the sota performance of Pyramid Fusion.

For the LiDAR-based setting, we use PointPillars (Lang et al., 2019) as the encoder and use the 64-channel LiDAR as input. The detection range is set to $x \in [-102.4m, +102.4m]$, $y \in [-102.4m, +102.4m]$. The voxelization grid size $[0.4m, 0.4m]$, and we further downsample the feature map by $2\times$ and shrink the feature dimension to 64 for message sharing. The multi-scale feature dimension of Pyramid Fusion is [64,128,256] and the ResNeXt layers have [3,5,8] blocks each. Foreground estimators are $1 \times 1$ convolution with channel [64,128,256]. The hyper-parameter $\alpha_\ell = \{0.4, 0.2, 0.1\}_{\ell=1,2,3}$.

For the camera-based setting, we use Lift-Splat (Philion & Fidler, 2020) with EfficientNet (Tan & Le, 2019) as the image encoder, and resize the image input to 384 px height. The detection range is set to $x \in [-51.2m, +51.2m]$, $y \in [-51.2m, +51.2m]$. The BEV feature map is initialized with size $[0.4m, 0.4m]$, downsampled by $2\times$ and shrinked to dimension 64 for message sharing. The hyperparameters of Pyramid Fusion is the same as the LiDAR-based model. Depth supervision is incorporated.

We present our collaborative perception results in Table. 5 and all methods are collectively trained in an end-to-end manner. Experiments show the superiority of our multiscale and foreground-aware fusion strategy.

Table 5: Comparison of the performance of different collaborative 3D object detection model in homogeneous setting on OPV2V and DAIR-V2X. Results show that our Pyramid Fusion outperforms the state-of-the-art methods in LiDAR-based homogeneous collaboration and camera-based homogeneous collaboration.

| Dataset | OPV2V | | | | DAIR-V2X | | | |
|---|---|---|---|---|---|---|---|---|
| Method | LiDAR-based | | Camera-based | | LiDAR-based | | Camera-based | |
| | AP50 ↑ | AP70 ↑ | AP50 ↑ | AP70 ↑ | AP30 ↑ | AP50 ↑ | AP30 ↑ | AP50 ↑ |
| No Collaboration | 0.782 | 0.634 | 0.405 | 0.216 | 0.421 | 0.405 | 0.014 | 0.004 |
| F-Cooper  Chen et al. (2019) | 0.763 | 0.481 | 0.469 | 0.219 | 0.723 | 0.620 | 0.115 | 0.026 |
| DiscoNet  Li et al. (2021) | 0.882 | 0.737 | 0.517 | 0.234 | 0.746 | 0.685 | 0.083 | 0.017 |
| AttFusion  Xu et al. (2022c) | 0.878 | 0.751 | 0.529 | 0.252 | 0.738 | 0.673 | 0.094 | 0.021 |
| V2XViT  Xu et al. (2022b) | 0.917 | 0.790 | 0.603 | 0.289 | 0.785 | 0.521 | 0.198 | 0.057 |
| CoBEVT  Xu et al. (2022a) | 0.935 | 0.821 | 0.571 | 0.261 | 0.787 | 0.692 | 0.182 | 0.042 |
| HM-ViT  Xiang et al. (2023) | 0.950 | 0.873 | 0.643 | 0.370 | 0.818 | 0.761 | 0.163 | 0.044 |
| Pyramid Fusion | **0.963** | **0.926** | **0.689** | **0.468** | **0.832** | **0.790** | **0.282** | **0.128** |

## A.4   OPV2V-H DATASET DETAILS

The existing collaborative perception benchmarks focus solely on a single sensor type. However, in real-world scenarios, agents often have varied sensor setups due to the different deployment timelines and expenses. To bridge this gap, we propose a heterogeneous dataset, OPV2V-H, where agents are set up with diverse configurations. OPV2V-H is collected with OpenCDA (Xu et al., 2021) and CARLA (Dosovitskiy et al., 2017)(under MIT license). Figure. 9 shows the simulation environment. OpenCDA provides the driving scenarios that ensure the agents drive smoothly and safely, including the vehicle's initial location and moving speed. CARLA provides the maps, and weather and controls the movements of the agents.

**Data collection.** The simulation settings and traffic patterns are derived from the playback of OPV2V, as referenced in (Xu et al., 2022c). Specifically, we position both the autonomous and background agents based on the standard configuration of OPV2V. Additionally, we outfit each autonomous agent with a broader range of sensors, totaling 11, including 4 cameras, 4 depth sensors,

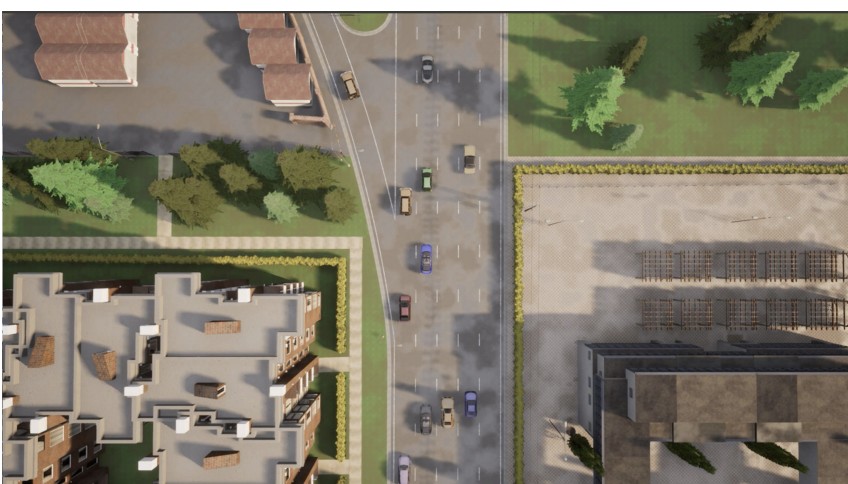

Figure 9: OPV2V-H is co-simulated by OpenCDA Xu et al. (2021) and CARLA Dosovitskiy et al. (2017).

and 3 LiDAR sensors. The cameras provide views from the front, left, right, and rear. The depth sensors are arranged in a similar fashion to the cameras. The LiDAR configurations comprise 64-channel, 32-channel, and 16-channel setups, which are typical in real-world situations. The sensor data and the sensor configurations are recorded during data collection.

**Sensor configuration.** The four RGB and depth cameras face forward, backward, right, and left with a yaw degree of $0°$, $100°$, $-100°$, and $180°$, respectively. And the corresponding offset to the agent center is $(2.5, 0.0, 1.0)$, $(0.0, 0.3, 1.8)$, $(0.0, -0.3, 1.8)$, and $(-2.0, 0.0, 1.5)$, respectively. The resolution of the captured image is $800 \times 600$. On each agent, all the cameras are fixed and their internal relative position and rotation degree are invariable. Camera intrinsics and extrinsic in global coordinates are recorded to support coordinate transformation across various agents. The 3 LiDAR sensors are attached at the same location on the agent with a rotation frequency of 20 Hz, the upper field of view 2, and lower field of view $-25$, and channels 16, 32, and 64, respectively, resulting in about $10k$, $20k$ and $40k$ point clouds, respectively. The translation (x, y, z) and rotation (w, x, y, z in quaternion) of each camera and LiDAR in both global and ego coordinates are recorded during data collection.

**Data statistics.** OPV2V-H provides fully annotated data, including synchronous images and point clouds with 3D bounding boxes of vehicles. In total, OPV2V-H has 10,524 samples, including 6374/1980/2170 in train/validation/test split, respectively. Each sample has 2-7 agents. Figure 10 shows captured sensor data of the same agent, including the 4 RGB images, 4 depth images from four views (front, left, right, back), and 3 LiDAR point clouds from the same view.

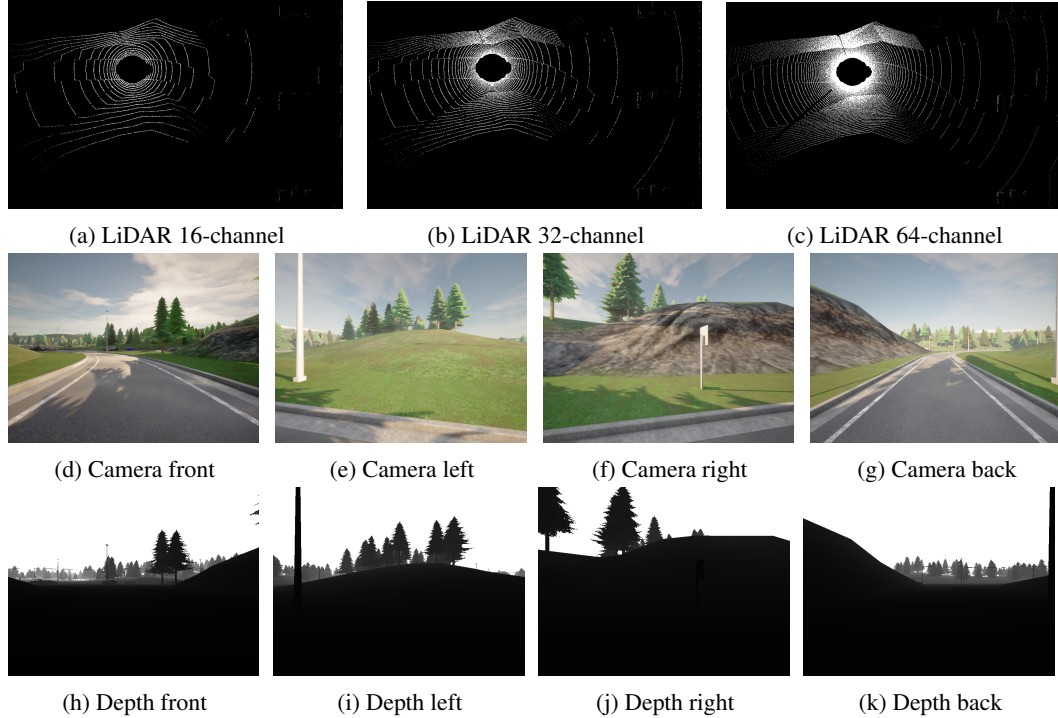

(a) LiDAR 16-channel      (b) LiDAR 32-channel      (c) LiDAR 64-channel

(d) Camera front    (e) Camera left    (f) Camera right    (g) Camera back

(h) Depth front    (i) Depth left    (j) Depth right    (k) Depth back

Figure 10: Each agent is equipped with 3 LiDAR, 4 cameras, and 4 depth sensors in OPV2V-H.

## A.5 FEATURE ENCODING DETAILS

**PointPillars Lang et al. (2019)** ($\mathbf{L}_P^{(x)}$ in the paper) divides the point cloud data into pillars, which are vertical columns of points. They are then converted to a stacked pillar tensor and pillar index tensor. The encoder uses the stacked pillars to learn a set of features that can be scattered back to a 2D pseudo-image for a convolutional neural network. This forms the BEV feature.

**SECOND Yan et al. (2018)** ($\mathbf{L}_S^{(x)}$ in the paper) begins by voxelizing the 3D point cloud data, creating a 3D grid of voxels. Each voxel contains the point cloud data within its boundaries. SECOND then uses sparse convolutions to process these voxels. Stacking the voxel features along the height dimension results in a BEV feature map.

**Lift-Splat Philion & Fidler (2020)** ($\mathbf{C}_E^{(x)}$ and $\mathbf{C}_R^{(x)}$ in the paper) represents a more recent and innovative approach for camera modality. It starts by "lifting" the image features from an image encoder (here $\mathbf{C}_E^{(x)}$ uses EfficientNet and $\mathbf{C}_R^{(x)}$ uses ResNet50) with a predicted depth distribution into 3D space. After this, it "splats" these features onto a 2D grid, effectively projecting features onto a BEV plane. In our paper, we use this feature projection mechanism with two different image encoders.

## A.6 TRAINING DETAILS

We introduce how we train the baseline model and our HEAL model on OPV2V-H dataset. The training procedure on DAIR-V2X dataset is the same.

**Baseline Model Training** Baseline intermediate fusion, including F-Cooper (Chen et al., 2019), AttFusion (Xu et al., 2022c), DiscoNet (Li et al., 2021), V2X-ViT (Xu et al., 2022b), CoBEVT (Xu et al., 2022a), HMViT (Xiang et al., 2023), will conduct a collectively end-to-end training to integrate a new agent type. Assume that when a new agent type seeks to join the existing collaboration, there are a total of $m$ distinct agent types in the scenario, inclusive of the new intelligent agent. During the training, we randomly select an agent from the scenario to serve as the ego agent and assign it an agent type from one of the $m$ distinct types with uniform probability. Concurrently, each of the other agent in the scene is assigned an agent type with the same uniform probability. All agents

encode their features according to the encoder specific to their type and transmit the information to the ego agent for feature fusion. During the validation phase, we prepared a JSON file in advance, which contains the pre-assigned agent types for each agent in the scene. Note that the agent types within the JSON file are also assigned with a uniform probability distribution (but since it is stored in the file, it keeps the same for each validation and for each model). We select a specific agent from the scenario to serve as the ego agent for computing the validation loss. In OPV2V-H, this would be the agent with the smallest agent ID.

The rationale behind this design is that during training, we can randomly permute the types of agents within the scenario and randomly select an agent to serve as the ego, thereby achieving data augmentation. Concurrently, during validation, the validation scenarios for all methods are consistent and fixed. This ensures an accurate reflection of whether the training has converged and facilitates a fair comparison between methods.

**HEAL Training** As stated in Sec. 4, the training of HEAL can be divided into collaboration base training and new agent type training. For collaboration base training, it is the same as Baseline Model Training which is trained collectively in an end-to-end manner. So the training strategy is also the same with a total agent type number $m = 1$. For new agent training, all agents in the scenario will be assigned to new agent type. However, because the training is only carried out on single agents, we will randomly select an agent from the scene as ego and use its sensor data for individual training. During the validation phase, we consistently select a specific agent within the scenario as the ego to compute the validation loss. In OPV2V-H, this would be the agent with the smallest agent ID.

## B    SPATIAL TRANSFORMATION

Each agent generates BEV features in its own coordinate system, and the ego agent requires spatial transformation to unify all coming features into its coordinate system. This step requires knowing the (i) relative pose ${}^i\xi_j \in \mathbb{R}^{4 \times 4}$ between vehicles and (ii) the physical size of the BEV feature $D_x$, $D_y$ in meters. We assume these meta information (global pose and BEV feature map physical size) will be sent in V2V communication. Note that relative pose can be calculated between global poses.

To perform the spatial transformation, we adopt the `affine_grid` and `grid_sample` functions from `torch.nn.functional` package.

Since the feature is in BEV space, we deprecate the z-axis component in ${}^i\xi_j$ and normalize it with $D_x$ and $D_y$. It results in a normalized and unitless affine matrix, which will be the input of `affine_grid` and `grid_sample` together with the feature map. With these two functions executed continuously, we will transform the feature map from other agent's ego perspective to our own coordinate. Just like the affine transformation of a 2D image.

