# OpenReview forum: "An Extensible Framework for Open Heterogeneous Collaborative Perception"
_ICLR.cc/2024/Conference — ICLR 2024 poster_

### Official Review · Reviewer_prVY · 2023-10-19

**Soundness:** 3 good
**Presentation:** 2 fair
**Contribution:** 2 fair
**Rating:** 8
**Confidence:** 5

**Summary:**

This article introduces a novel problem about collaborative perception, which is extensible heterogeneous agent collaborative perception. To address this problem, a novel extensible collaborative perception framework.is proposed, which includes a novel Pyramid Fusion method. In addition, a new large-scale database is proposed based on OPV2V. The introduced problem is interesting and has significance in engineering. However, the explanation of certain content is not clear enough.

**Strengths:**

1. The introduced extensible heterogeneous agent collaborative perception problem is interesting and has significance in engineering.
2. The proposed HEAL framework is novel and holds state-of-the-art performance.
3. A new large-scare database is proposed to facilitate the research of heterogeneous collaborative perception.

**Weaknesses:**

1. The content of the Section 3 is somewhat limited, please make some extensions or merge it with other sections.
2. The article has not mentioned how to extract features from different modalities and convert them into the required BEV features.
3. The article has not mentioned how to perform spatial transformation and alignment of the features from heterogeneous agents in different coordinate systems. Please clarify this.
4. During the training of the new-type agent, the new agent seems not to get collaborative information from other agents, which may have an impact on perception performance. This should be explained.
5. On the other hand, the Pyramid Fusion module and detection head haven’t been affected by new type agents. It means that new-type agents did not participate in collaborative perception during the training process. This could be better clarified.
6. The claim of the performance of the proposed Pyramid Fusion should be supported by experiment results and observations, please supplement relevant ablation experiments.

Sincerely,

**Questions:**

1. What’s the structure of the encoder in the proposed framework?
2. Will the number of input channels for the Pyramid Fusion module change when a new agent is added?
3. Why can the performance of this method be improved so much on AP70?

Sincerely,

---

> ### Author Response · Authors · 2023-11-19
> **Response #1**
>
> Thank you for your thorough review and valuable feedback on this paper. Your insights are greatly appreciated and will guide my revisions. Please let me answer your confusion in the following paragraphs.
>
> > **W1:** The content of the Section 3 is somewhat limited, please make some extensions or merge it with other sections.
>
> Thanks for your advice! The proposed open heterogeneous collaborative perception problem holds significant practical significance, and therefore, it warrants a individual section to introduce it. In the revision, we have further expanded upon its content to provide a more comprehensive introduction to this highly practical and novel challenge.
>
> > **W2:** The article has not mentioned how to extract features from different modalities and convert them into the required BEV features.
>
> Thanks for pointing it out. The feature encoding methods are as follows:
>
> 1. LiDAR modality
>
> - **PointPillars[1]** ($\mathbf{L}^{(x)}_\mathit{P}$ in the paper) divides the point cloud data into pillars, which are vertical columns of points. They are then converted to a stacked pillar tensor and pillar index tensor. The encoder uses the stacked pillars to learn a set of features that can be scattered back to a 2D pseudo-image for a convolutional neural network. This forms the BEV feature.
>
> - **SECOND[2]** ($\mathbf{L}^{(x)}_\mathit{S}$ in the paper) begins by voxelizing the 3D point cloud data, creating a 3D grid of voxels. Each voxel contains the point cloud data within its boundaries. SECOND then uses sparse convolutions to process these voxels. Stacking the voxel features along the height dimension results in a BEV feature map.
>
> 2. Camera modality
>
> - **Lift-Splat[3]** ($\mathbf{C}^{(x)}_E$ and $\mathbf{C}^{(x)}_R$ in the paper) represents a more recent and innovative approach for camera modality. It starts by "lifting" the image features from an image encoder (here $\mathbf{C}^{(x)}_E$ uses EfficientNet and $\mathbf{C}^{(x)}_R$ uses ResNet50) with a predicted depth distribution into 3D space. After this, it "splats" these features onto a 2D grid, effectively projecting features onto a BEV plane. In our paper, we use this feature projection mechanism with two different image encoders.
>
> [1] Fast Encoders for Object Detection from Point Clouds. CVPR 2019.
>
> [2] SECOND: Sparsely Embedded Convolutional Detection. Sensors 2018.
>
> [3] Lift, Splat, Shoot: Encoding Images From Arbitrary Camera Rigs by Implicitly Unprojecting to 3D. ECCV 2020.
>
> > **W3:** The article has not mentioned how to perform spatial transformation and alignment of the features from heterogeneous agents in different coordinate systems. Please clarify this.
>
> Since agent generates BEV features in its own coordinate system, the ego agent requires spatial transformation to unify all coming features into its own coordinate system. This step requires knowing the (i) relative pose $^i\xi_j \in \mathbb{R}^{4\times 4}$ between vehicles and (ii) the physical size of the BEV feature $D_x$, $D_y$ in meters. We assume these meta information (global pose and BEV feature map physical size) will be sent in V2V communication. Note that relative pose can be calculated between global poses.
>
> To perform the spatial transformation, we adopt the ```affine_grid``` and ```grid_sample``` functions from ```torch.nn.functional``` package.
>
> Since the feature is in BEV space, we deprecate the z-axis component in $^i\xi_j$ and normalize it with $D_x$ and $D_y$. It results in a normalized and unitless affine matrix, which will be the input of ```affine_grid``` and ```grid_sample``` together with the feature map. With these two functions executed continuously, we will transform the feature map from the other agent's ego perspective to our own coordinate, just like the affine transformation of a 2D image.
>
> We also supplement this in the Appendix.

---

> ### Author Response · Authors · 2023-11-19
> **Response #2**
>
> > **W4:** During the training of the new-type agent, the new agent seems not to get collaborative information from other agents, which may have an impact on perception performance. This should be explained.
>
> As the variety of agent types in the scene grows, training without the use of collaborative information does not exhibit drawbacks; instead, it has a beneficial impact.
>
> This is because collaborative training may result in unbalanced and insufficient training in situations with numerous agent types. Consider this scenario: in the OPV2V-H dataset, each scene averages 3 agents, and our experiment involves 4 types of agents. Given the nature of the training data, such collective training is prone to unbalanced training. This remains the case despite our implementation of extensive data augmentation and setting more epochs to promote convergence.
>
> We have conducted relevant experiments on OPV2V-H as follows:
>
> | Infer $\mathbf{L}_p^{(64)}$ +$\mathbf{C}^{(384)}_E$ +$\mathbf{L}^{(32)}_S$ + $\mathbf{C}^{(336)}_R$ on OPV2V-H | AP30 | AP50 | AP70 |
> |----------------------------------------------------------------------------------------------------------------|------|------|------|
> | Collective Training + Pyramid Fusion | 0.899 | 0.881 | 0.769 |
> | Back. Align. Training + Pyramid Fusion (HEAL) | **0.908** | **0.894** | **0.813** |
>
> We see that, when we use collective training for newly added heterogeneous agent types, the performance is suboptimal to single-agent training based on backward alignment. Backward alignment does not involve collaboration, thereby circumventing the issues of training imbalance and insufficiency in collaborative training. Individual training enables new agents to converge more rapidly, aligning them to a unified feature space for collaboration. In real situations, the number of agent types is always growing, and it becomes increasingly difficult to collect enough spatio-temporally synchronized multi-agent data including types. Therefore, training on a single agent is an advantage. Therefore, the advantages demonstrated by training on a single agent are even more significant.
>
> > **W5:** On the other hand, the Pyramid Fusion module and detection head haven’t been affected by new type agents. It means that new-type agents did not participate in collaborative perception during the training process. This could be better clarified.
>
>
> Thank you for your suggestion. We would like to clarify that new-type agent training does not involve any collaborative agents. This important detail will be reiterated in the main text for emphasis.
>
>
> > **W6:** The claim of the performance of the proposed Pyramid Fusion should be supported by experiment results and observations, please supplement relevant ablation experiments.
>
> We provide a detailed ablation study on Pyramid Fusion on OPV2V-H dataset as follows. The same as our experiment setting, we gradually add $\mathbf{C}^{(384)}_E$, $\mathbf{L}^{(32)}_S$, $\mathbf{C}^{(336)}_R$ agents to the scene starting with one  $\mathbf{L}_p^{(64)}$ agent.
>
>
> | AP50 | - | - | - | - | - | - | - |
> |------|--|--|--|--|--|--|--|
> | Foreground Estimator | Foreground Estimator  Supervision | Multiscale Structure | Encoder Layer | $\mathbf{L}_p^{(64)}$ | $\mathbf{L}_p^{(64)}$ +$\mathbf{C}^{(384)}_E$ | $\mathbf{L}_p^{(64)}$ +$\mathbf{C}^{(384)}_E$ +$\mathbf{L}^{(32)}_S$ | $\mathbf{L}_p^{(64)}$ +$\mathbf{C}^{(384)}_E$ +$\mathbf{L}^{(32)}_S$ +$\mathbf{C}^{(336)}_R$ |
> | x | x | x | ResNeXt | 0.744 | 0.794 | 0.856 | 0.857 |
> | x | x | √ | ResNeXt | 0.754 | 0.802 | 0.857 | 0.855 |
> | √ | √ | x | ResNeXt | 0.753 | 0.82 | 0.878 | 0.879 |
> | √ | x | √ | ResNeXt | 0.745 | 0.799 | 0.87 | 0.864 |
> | √ | √ | √ | vanilla CNN | 0.757 | 0.823 | 0.879 | 0.88 |
> | √ | √ | √ | ResNeXt | **0.764** | **0.826** | **0.892** | **0.894** |
> | **AP70** | - | - | - | - | - | - | - |
> | Foreground Estimator | Foreground Estimator  Supervision | Multiscale Structure | Encoder Layer | $\mathbf{L}_p^{(64)}$ | $\mathbf{L}_p^{(64)}$ +$\mathbf{C}^{(384)}_E$ | $\mathbf{L}_p^{(64)}$ +$\mathbf{C}^{(384)}_E$ +$\mathbf{L}^{(32)}_S$ | $\mathbf{L}_p^{(64)}$ +$\mathbf{C}^{(384)}_E$ +$\mathbf{L}^{(32)}_S$ +$\mathbf{C}^{(336)}_R$ |
> | x | x | x | ResNeXt | 0.618 | 0.674 | 0.763 | 0.761 |
> | x | x | √ | ResNeXt | 0.642 | 0.692 | 0.774 | 0.772 |
> | √ | √ | x | ResNeXt | 0.633 | 0.699 | 0.785 | 0.782 |
> | √ | x | √ | ResNeXt | 0.628 | 0.689 | 0.786 | 0.781 |
> | √ | √ | √ | vanilla CNN | 0.65 | 0.723 | 0.806 | 0.807 |
> | √ | √ | √ | ResNeXt | **0.662** | **0.726** | **0.812** | **0.813** |
>
>
>
> It shows that all the component are beneficial to the performance gain.

---

> ### Author Response · Authors · 2023-11-19
> **Response #3**
>
> > **Q1:** What’s the structure of the encoder in the proposed framework?
>
>
>
> 1. LiDAR modality
>
> - $\mathbf{L}^{(x)}_\mathit{P}$ refers to PointPillar[1].
>
> - $\mathbf{L}^{(x)}_\mathit{S}$ refers to SECOND[2].
>
> 2. Camera modality
>
> - $\mathbf{C}^{(x)}_\mathit{E}$ refers to Lift-Splat[3] with EfficientNet as image encoder.
>
> - $\mathbf{C}^{(x)}_\mathit{R}$ refers to Lift-Splat[3] with ResNet50 as image encoder.
>
> We introduced them in the response to **W2**.
>
> > **Q2:** Will the number of input channels for the Pyramid Fusion module change when a new agent is added?
>
> The input feature channel for the Pyramid Fusion module will not change. While a new agent's model may encode features with a different channel number, we can append some additional network after the encoder to adjust the channel of features. The added network layers will be trained with the encoder together to achieve optimal collaboration performance.
>
>
> > **Q3:** Why can the performance of this method be improved so much on AP70?
>
> Thanks for your insightful comment and this is a key question. We clarify that the performance gain comes from both the training strategy and fusion method are influencing factors, while **the training strategy based on backward alignment is more critical**.
>
> This is because, especially when there are many agent types, such as 4 four heterogeneous agent types shown in the paper, it is difficult to balance their training in collective training. In the OPV2V-H dataset, the average number of agents in the scene is only close to 3. The collective training method can easily fall into insufficient training for multiple agent types (for example, the 4 types in our paper). Moreover, the pre-training techniques employed in the HM-ViT paper are also insufficient in addressing the problem of imbalance, significantly when the number of agent types surpasses the number of agents present in the scene.
>
> Our novel backward alignment does not require collaboration, thereby eliminating the issues of training imbalance and insufficiency of each agent type in collective training. It enables new agents have easier convergence and easier training with alignment to a unified feature space for collaboration. This is very important for precise detection, which is reflected in the AP70 metric.
>
>
> | Infer $\mathbf{L}_p^{(64)}$ +$\mathbf{C}^{(384)}_E$ +$\mathbf{L}^{(32)}_S$ + $\mathbf{C}^{(336)}_R$ on OPV2V-H | AP30 | AP50 | AP70 |
> |----------------------------------------------------------------------------------------------------------------|------|------|------|
> | Collective Training + HM-ViT | 0.891 | 0.876 | 0.755 |
> | Collective Training + Pyramid Fusion | 0.899 | 0.881 | 0.769 |
> | Back. Align. Training + Pyramid Fusion (HEAL) | **0.908** | **0.894** | **0.813** |
>
>
> From the table above, we see that "Collective Training + Pyramid Fusion" exhibits a slightly higher AP performance with some baseline methods. Thus, it demonstrates significant performance improvements in AP70 come from backward alignment.
>
> References:
>
> [1] Fast Encoders for Object Detection from Point Clouds. CVPR 2019.
>
> [2] SECOND: Sparsely Embedded Convolutional Detection. Sensors 2018.
>
> [3] Lift, Splat, Shoot: Encoding Images From Arbitrary Camera Rigs by Implicitly Unprojecting to 3D. ECCV 2020.

---

> ### Comment · Reviewer_prVY · 2023-11-20
> **Comment**
>
> The reviewer would like to thank the authors for their responses with added analyses and results, which helped solve many concerns. The authors clarify that collaborative training may result in unbalanced and insufficient training in situations with numerous agent types, which is an interesting observation and could lead to further research in this line.
>
> The reviewer would like to elevate the rating. The added analyses should be incorporated into the camera-ready version.
>
> Sincerely,

---

### Official Review · Reviewer_bqtt · 2023-10-28

**Soundness:** 3 good
**Presentation:** 3 good
**Contribution:** 2 fair
**Rating:** 5
**Confidence:** 5

**Summary:**

This work presents a novel training schema and fusion network for heterogeneous cooperative perception where agents may have different kinds of modalities/sensor configurations. The proposed framework aligns the BEV features from different modalities to the common feature space and fuses them to reason 3D object detections.

**Strengths:**

* The proposed training design is simple yet effective for faster convergence and higher performance.
* The proposed methods achieve outstanding performance.

**Weaknesses:**

* When comparing with other SOTA algorithms, is the same training strategy used for a fair comparison? Is the main performance boost from the training strategy or the multi-modal fusion design?
* The fusion model design (except the residual part) shows similarity with existing methods like who2com/where2comm/disconet. Please justify and highlight the differences and novelty. Please also benchmark the performance under the same training strategy with only different fusion networks so as to demonstrate the effectiveness of the proposed fusion module.
* How to ensure each modality can be aligned to the common feature space? For example, it may be hard to extract the 3D features
from camera/radar data which are expected to be as good as (aligned) as the LiDAR features. From 6, we can see that camera bev features are more vague than LiDAR features, which could not show the aligned effect. Please justify this design choice.
* Can the network scale to different sensor modality combinations? For example, changing/fixing the ego modality with dynamic collaborator sensor modalities. What is the sensitivity of the network with respect to this sensor modality combination ratio?
* The 5 hour training time is for single modality. What is the overall training time and the associated time for the compared model?

**Questions:**

* The author argued that "late fusion is suboptimal due to the communication latency". However, as shown in DiscoNet/V2X-ViT/V2VNet etc., intermediate fusion methods usually require larger bandwidth requirements than late fusion, which can even lead to potentially larger communication latency compared with late fusion.
* How the result of HM-ViT is reproduced? Is the heterogeneity used for all 4 modalities? Or only two modalities are used as the original paper?
* What is the inference time of the proposed method?
* What is the influence of the modality choice in the base collaboration training?

---

> ### Author Response · Authors · 2023-11-19
> **Response #1**
>
> Thank you very much for dedicating your time and expertise to review our submission. The depth of your analysis and the constructive criticism you have provided are immensely beneficial. I sincerely hope my reply can answer your questions.
>
> > **W1:** (a) When comparing with other SOTA algorithms, is the same training strategy used for a fair comparison? (b) Is the main performance boost from the training strategy or the multi-modal fusion design?
>
>
> (a) For the first question, given that HEAL represents a framework-level innovation with the novel training strategy, it is justifiable to conduct the comparsion. Additionally, for a fair assessment of the fusion module, we performed an extra experiment. In this experiment, we replaced our backward alignment based training with the collective training approach used in the baseline. The results are as follows:
>
> | Infer $\mathbf{L}_p^{(64)}$ +$\mathbf{C}^{(384)}_E$ +$\mathbf{L}^{(32)}_S$ + $\mathbf{C}^{(336)}_R$ on OPV2V-H | AP30 | AP50 | AP70 |
> |----------------------------------------------------------------------------------------------------------------|------|------|------|
> | Collective Training + HM-ViT | 0.891 | 0.876 | 0.755 |
> | Collective Training + Pyramid Fusion | 0.899 | 0.881 | 0.769 |
> | Back. Align. Trainig + Pyramid Fusion **(HEAL)** | **0.908** | **0.894** | **0.813** |
>
> We see that under the same training strategy, our fusion module slightly surpasses existing SOTA method HM-ViT.
>
> Furthermore, we show the benchmark of pure LiDAR-based collaboration and pure Camera-based collaboration in Appendix. In the table (from Appendix) below , methods are also trained with the same strategy. Some factors, such as whether all detections are sorted by confidence before AP calculating and detection range setting, may lead to AP differences with other papers, but we got the same SOTA method ranking as in the HM-ViT paper. We believe it is a very fair comparison:
>
> | Dataset | OPV2V | - | - | - | DAIR-V2X | - | - | - |
> |---------|-------|--|--|--|----------|--|--|--|
> | **Method** | LiDAR-based | LiDAR-based | Camera-based | Camera-based | LiDAR-based | LiDAR-based | Camera-based | Camera-based |
> | - | AP50 | AP70 | AP50 | AP70 | AP30 | AP50 | AP30 | AP50 |
> | No Fusion | 0.782 | 0.634 | 0.405 | 0.216 | 0.421 | 0.405 | 0.014 | 0.004 |
> | F-Cooper | 0.821 | 0.632 | 0.469 | 0.219 | 0.723 | 0.620 | 0.115 | 0.026 |
> | DiscoNet | 0.882 | 0.737 | 0.517 | 0.234 | 0.746 | 0.685 | 0.083 | 0.017 |
> | AttFusion | 0.878 | 0.751 | 0.529 | 0.252 | 0.738 | 0.673 | 0.094 | 0.021 |
> | V2XViT | 0.917 | 0.790 | 0.603 | 0.289 | 0.785 | 0.521 | 0.198 | 0.057 |
> | CoBEVT | 0.935 | 0.821 | 0.571 | 0.261 | 0.787 | 0.692 | 0.182 | 0.042 |
> | HMViT | 0.950 | 0.873 | 0.643 | 0.370 | 0.818 | 0.761 | 0.163 | 0.044 |
> | Pyramid Fusion | **0.963** | **0.926** | **0.689** | **0.468** | **0.832** | **0.790** | **0.282** | **0.128** |
>
>
>
> (b) For the second question, both the training strategy and the fusion method are key components, while the training strategy based on backward alignment is more critical. This is because, especially when there are many agent types, such as 4 four heterogeneous agent types shown in the paper, it is actually difficult to balance their training in collective training, **even with pre-train.** In the OPV2V-H dataset, the average number of agents in the scene is only close to 3, while we have 4 agent types. The collective training method can easily fall into insufficient training for each heterogeneous agent's collaboration. Moreover, the pre-training techniques employed in the HM-ViT paper are also inadequate in addressing the problem of imbalance, especially when the number of agent types surpasses the number of agents present in the training data.
>
> Conversely, backward alignment does not involve collaboration, thereby circumventing the issues of training imbalance and insufficiency in collaborative training. Individual training enables new agents to converge more rapidly, aligning them to a unified feature space for collaboration. When we use collective training + Pyramid Fusion, the same training strategy with baselines, to train newly added heterogeneous agent types, the performance is also suboptimal to HEAL with backward alignment. Here are the experimental results.
>
>
>
> | Infer $\mathbf{L}_p^{(64)}$ +$\mathbf{C}^{(384)}_E$ +$\mathbf{L}^{(32)}_S$ + $\mathbf{C}^{(336)}_R$ on OPV2V-H | AP30 | AP50 | AP70 |
> |----------------------------------------------------------------------------------------------------------------|------|------|------|
> | Collective Trainig + Pyramid Fusion | 0.899 | 0.881 | 0.769 |
> | Back. Align. Trainig + Pyramid Fusion **(HEAL)** | **0.908** | **0.894** | **0.813** |
>
>
>
> This demonstrates significant performance improvements from backward alignment.

---

> ### Author Response · Authors · 2023-11-19
> **Response #2**
>
> > **W2:** The fusion model design (except the residual part) shows similarity with existing methods like who2com/where2comm/disconet. Please justify and highlight the differences and novelty. Please also benchmark the performance under the same training strategy with only different fusion networks to demonstrate the effectiveness of the proposed fusion module.
>
> The proprietary design of Pyramid Fusion renders it uniquely compatible with the backward alignment training strategy. Not every fusion algorithm can work with our backward alignment. To simplify the explanation, we conceptualize collaborative perception as a graph structure, where the nodes represent the features of different agents, and the edges signify the interactions between these agents. It should be clarified that not all fusion methods can work with backward alignment, especially the algorithms that need pairwise edge interaction.
>
> The following are the differences and novelty designs by Pyramid Fusion.
>
> 1. **Node level weight prediction**
> During the training of new agents, Pyramid Fusion does not require interaction with the features of other agents to predict an edge weight collectively. Instead, its weight is node-level, only derived from its foreground estimation. This modular design and independence are well-suited and more appropriate when no collaboration is involved in new agent training.
> As for DiscoNet/who2com/Where2comm, the weighting mechanisms are derived from pairwise interactions. For instance, DiscoNet and who2com concatenate ego and neighbor features to predict edge weights, whereas where2comm employs an attention mechanism with learnable parameters to calculate pairwise weights. A critical issue arises when training new agents alone without collaboration, as these methods are limited to one self-loop at the single-agent node on the collaboration graph.  For a self-loop input, a softmax function invariably outputs a weight of 1. This results in their edge weight prediction being ineffective during backward alignment, and new agent types do not adapt to the parameters of edge weight prediction. Consequently, during actual inference, the estimated edge weights between heterogeneous agents are erroneous, preventing the agents from collaborating effectively.
>
>
> 2. **Explicit foreground estimators**
> In Pyramid Fusion, we have designed explicit foreground estimators and supervision for the predicted foreground maps. The foreground estimators predict the foreground probability for each BEV pixel for effective feature fusion. This supervision serves two crucial purposes: (i) It promotes the learning of foreground estimators, leading to more accurate weight predictions. (ii) More significantly, they offer feature alignment supervision signals during backward alignment. Since the parameters of the foreground estimator are also fixed for new heterogenous agent training, they go through the same individual supervision on foreground estimation, which encourages the encoder to produce features that align with those of the base agent.
> For DiscoNet/who2com/Where2comm, the absence of an estimator and supervision on weight prediction leads to suboptimal fusion and poor feature alignment.
>
>
> 3. **Multiscale network structure**
> Our approach with multiscale feature fusion and supervision further facilitates the fusion. Particularly the multiscale foreground estimator, which, through supervision at varying granularities, ensures the alignment of features between new agents and the base agent during backward alignment. This aspect of our methodology ensures that the new agents are effectively aligned with the base agents.
> For DiscoNet/who2com/Where2comm, they only employ a single-scale network structure, which is suboptimal to our multiscale design in feature learning and fusion.
>
> We benchmark their performance under the same training strategy. As we mentioned previously, backward alignment is unsuitable for feature fusion algorithms with edge interaction. Thus, with the same training strategy, HEAL significantly outperforms other methods. Experiments on OPV2V-H dataset:
>
> | AP50 | - | - | - | - |
> |------|--|--|--|--|
> | Fusion Backbone | $\mathbf{L}_p^{(64)}$ | $\mathbf{L}_p^{(64)}$ +$\mathbf{C}^{(384)}_E$ | $\mathbf{L}_p^{(64)}$ +$\mathbf{C}^{(384)}_E$ +$\mathbf{L}^{(32)}_S$ | $\mathbf{L}_p^{(64)}$ +$\mathbf{C}^{(384)}_E$ +$\mathbf{L}^{(32)}_S$ +$\mathbf{C}^{(336)}_R$ |
> | who2com | 0.742 | 0.09 | 0.016 | 0.017 |
> | Where2comm | 0.743 | 0.773 | 0.774 | 0.771 |
> | Ours | **0.764** | **0.826** | **0.892** | **0.894** |
> | **AP70** | - | - | - | - |
> | Fusion Backbone | $\mathbf{L}_p^{(64)}$ | $\mathbf{L}_p^{(64)}$ +$\mathbf{C}^{(384)}_E$ | $\mathbf{L}_p^{(64)}$ +$\mathbf{C}^{(384)}_E$ +$\mathbf{L}^{(32)}_S$ | $\mathbf{L}_p^{(64)}$ +$\mathbf{C}^{(384)}_E$ +$\mathbf{L}^{(32)}_S$ +$\mathbf{C}^{(336)}_R$ |
> | who2com | 0.621 | 0.05 | 0.006 | 0.006 |
> | Where2comm | 0.639 | 0.674 | 0.679 | 0.678 |
> | Ours | **0.662** | **0.726** | **0.812** | **0.813** |

---

> > ### Comment · Reviewer_bqtt · 2023-11-21
> >
> > For W1
> >
> > 1) the Node level weight prediction: it is still due to the proposed training strategy not the novel architecture/model-level design.
> >
> > 2) Explicit foreground estimators: Why Where2comm etc. lack an estimator and supervision? Where2comm uses the detection head (generator in equation (1) in the original where2comm paper) to generate the spatial confidence map. This detection head is trained and supervised with the training data as it is also used for generating the bounding box proposals. Please highlight the difference.

---

> > > ### Comment · Reviewer_bqtt · 2023-11-21
> > >
> > > For Q1, I am still not convinced that intermediate fusion has less bandwidth requirements than the late fusion. Though intermediate fusion can compress the features along the channel dimension, yet late fusion transmitts far less data as shown in works like OPV2V (Fig. 10 late fusion only requires 0.003MB data while even under 4096x compression, the intermediate still requires 0.016MB). Moreover, for example, if the detected bounding boxes only have 10 objects, then a matrix of size 10x7 need to be transmitted (7--x,y,z,dx,dy,dz,yaw).

---

> > > > ### Comment · Reviewer_bqtt · 2023-11-21
> > > >
> > > > I would encourage the author to add the inference time/training time comparison table in the revised paper. And please rephrase the comparison like "Training costs 5 hours for each new agent’s training, while HM-ViT takes more than 1 day to converge" as they use different comparison units. Please compare these two methods fairly using the same units.

---

> > > > > ### Author Response · Authors · 2023-11-22
> > > > >
> > > > > > I would encourage the author to add the inference time/training time comparison table in the revised paper. And please rephrase the comparison like "Training costs 5 hours for each new agent’s training, while HM-ViT takes more than 1 day to converge" as they use different comparison units. Please compare these two methods fairly using the same units.
> > > > >
> > > > > Thanks for your suggestion! We have made the comparison in the same units in our revision.

---

> > > > ### Author Response · Authors · 2023-11-22
> > > >
> > > > > For Q1, I am still not convinced that intermediate fusion has less bandwidth requirements than the late fusion. Though intermediate fusion can compress the features along the channel dimension, yet late fusion transmitts far less data as shown in works like OPV2V (Fig. 10 late fusion only requires 0.003MB data while even under 4096x compression, the intermediate still requires 0.016MB). Moreover, for example, if the detected bounding boxes only have 10 objects, then a matrix of size 10x7 need to be transmitted (7--x,y,z,dx,dy,dz,yaw).
> > > >
> > > > 1. Compressing  feature channel can reduce communication latency to a fully acceptable value. For your example from OPV2V, late fusion requires 0.003MB data and the intermediate fusion with compression requires 0.016 MB data. Although there is a gap of about 5 times between them, since their magnitudes are both very small, this will not be reflected as a significant gap in the communication delay. In the section of  "Effect of Compression Rates" in OPV2V, the authors assume a data broadcasting rate at 27 Mbps. Thus, the communication delay for 0.016MB data (intermediate fusion) is 5ms, which is totally acceptable.
> > > >
> > > > > Based on the V2V communication protocol [32], data broadcasting can achieve 27 Mbps at the range of 300 m. This represents that the time delay to deliver the message with a 4096x compression rate is only about 5 ms.
> > > > >
> > > > > [32] F. Arena and G. Pau, “An overview of vehicular communications,” Future Internet, vol. 11, no. 2, p. 27, 2019.
> > > >
> > > > 2. Compressing features in the spatial dimension can further reduce the training cost. As shown by Where2comm, feature maps can be very sparse, with very few features actually contributing to foreground detection.  We can transmit the most important features to further reduce the communication cost.
> > > > 3. In addition to compressing the communication volume to reduce communication delay, a recent work called CoBEVFlow[1] exhibits an effective solution to overcome communication delay. By explicitly estimating a BEV flow using bounding box information, the algorithm can compensate for the feature misalignment problem caused by communication delays.
> > > >
> > > > 4. As we have clarified before, this is not the focus of our research. There are a large number of papers exploring how to reduce the training cost and resist the communication delay, and we have already conducted relevant experiments to verify the feasibility of feature compression in HEAL. Our research focuses on open heterogeneous collaborative perception, that is, how to allow new heterogeneous agents to join the collaboration with high performance and low cost. Not every article is like Where2comm, focusing on reducing communication cost to the extreme. Those notable works, like V2X-ViT and HM-ViT, continue to hold their meaning and significance, even though the volume of communication they handle goes beyond that of late fusion.
> > > >
> > > > [1] Asynchrony-Robust Collaborative Perception via Bird's Eye View Flow, NeurIPS 2023.

---

> ### Author Response · Authors · 2023-11-19
> **Response #3**
>
> >**W3:** How to ensure each modality can be aligned to the common feature space? For example, it may be hard to extract the 3D features from camera/radar data which are expected to be as good as (aligned) as the LiDAR features. From 6, we can see that camera bev features are more vague than LiDAR features, which could not show the aligned effect. Please justify this design choice.
>
>
> We ensure that the features from the LiDAR agents and camera agents are aligned. We substantiate it from qualitative and quantitative aspects.
>
> **For qualitative analysis:**
>
> It should be clarified that, feature alignment means similar numerical intensity when describing the scene object, and it is reflected as similar brightness response in heat map visualization (as our figure 6 does).  The blurring of camera BEV features does not mean the misalignment in the feature space. Actually, it comes from the fact that the camera data has no scale information and cannot accurately restore the 3D scene like LiDAR. This is reflected in many camera-based detection paper like [1][2].
>
> In the Appendix of our revision, we have supplement more visualizations, including agent features at different scales. These examples consistently show similar feature responses in the foreground/background regions. In our visualization, both camera and LiDAR features exhibit similar and distinct delineations between foreground and background areas, demonstrating their alignment in the unified feature space.
>
> **For quantitative analysis:**
>
> We experiment to measure their feature similarity. We adopt the LiDAR encoder and camera encoder on the same ego agent (to ensure accurate spatial alignment) and calculate the cosine similarity between the LiDAR feature and camera feature. We evaluate on the whole testset, and the results on OPV2V-H dataset are as follows:
>
>
> | LiDAR and Camera Feature before fusion | Average Cosine Simiarlity |
> |----------------------------------------|---------------------------|
> | $\mathbf{F}_{j\rightarrow i}^{(1)}$ | 0.617 |
> | $\mathbf{F}_{j\rightarrow i}^{(2)}$ | 0.650 |
> | $\mathbf{F}_{j\rightarrow i}^{(3)}$ | 0.774 |
>
>
> It is worth noticing that many existing works[3,4] have also considered aligning the BEV features from LiDAR and cameras to the same space to facilitate multi-modality perception. X-Align[3] uses a Cross-Modal Feature Alignment loss to align LiDAR and camera BEV features for more accurate BEV segmentation. BEVDistill[4] uses a dense feature distillation to promote the similarity between the camera feature and the LiDAR feature. These papers all demonstrated that such a design is reasonable and results in considerable performance enhancements.
>
> [1] Categorical Depth Distribution Network for Monocular 3D Object Detection. CVPR 2021.
>
> [2] Lift, Splat, Shoot: Encoding Images From Arbitrary Camera Rigs by Implicitly Unprojecting to 3D. ECCV 2020.
>
> [3] X-Align: Cross-Modal Cross-View Alignment for Bird's-Eye-View Segmentation, WACV 2023.
>
> [4] BEVDistill: Cross-Modal BEV Distillation for Multi-View 3D Object Detection, ICLR 2023.

---

> ### Author Response · Authors · 2023-11-19
> **Response #4**
>
> >**W4:** Can the network scale to different sensor modality combinations? For example, changing/fixing the ego modality with dynamic collaborator sensor modalities. What is the sensitivity of the network with respect to this sensor modality combination ratio?
>
> HEAL is able to scale to different sensor modality combinations. Our training strategy inherently accounts for extreme LiDAR/camera ratio, because the new agents are trained independently (for example camera agent is trained without LiDAR collaborator). We conduct the LiDAR-Camera ratio experiment on OPV2V-H dataset as follows. The notation $\mathbf{L}_p^{(64)}$ + 3 $\mathbf{C}^{(384)}_E$ means ego agent is $\mathbf{L}_p^{(64)}$ and the collaborators are 3 $\mathbf{C}^{(384)}_E$ agents.
>
>
> | Modality Ratio Ablation (AP50) | $\mathbf{L}_p^{(64)}$ + 3 $\mathbf{L}_p^{(64)}$ | $\mathbf{L}_p^{(64)}$ + 3 $\mathbf{C}^{(384)}_E$ | $\mathbf{C}^{(384)}_E$ + 3 $\mathbf{L}_p^{(64)}$ | $\mathbf{C}^{(384)}_E$ + 3 $\mathbf{C}^{(384)}_E$ |
> |--------------------------------|-------------------------------------------------|--------------------------------------------------|--------------------------------------------------|---------------------------------------------------|
> | V2XViT | 0.932 | 0.846 | 0.855 | 0.452 |
> | CoBEVT | 0.919 | 0.849 | 0.727 | 0.432 |
> | HM-ViT | 0.927 | 0.838 | 0.864 | 0.435 |
> | HEAL | **0.954** | **0.848** | **0.87** | **0.458** |
> | **Modality Ratio Ablation (AP70)** | **$\mathbf{L}_p^{(64)}$ + 3 $\mathbf{L}_p^{(64)}$** | **$\mathbf{L}_p^{(64)}$ + 3 $\mathbf{C}^{(384)}_E$** | **$\mathbf{C}^{(384)}_E$ + 3 $\mathbf{L}_p^{(64)}$** | **$\mathbf{C}^{(384)}_E$ + 3 $\mathbf{C}^{(384)}_E$** |
> | V2XViT | 0.802 | 0.677 | 0.549 | 0.198 |
> | CoBEVT | 0.775 | 0.715 | 0.419 | 0.218 |
> | HM-ViT | 0.839 | 0.689 | 0.692 | 0.220 |
> | HEAL | **0.918** | **0.748** | **0.799** | **0.267** |
>
>
> We see that HEAL maintained commendable performance levels on the OPV2V-H dataset with different LiDAR-to-camera ratios. This showcases the resilience and adaptability of our approach in handling varying proportions of LiDAR and camera agents, ensuring consistent effectiveness across a range of modality ratios.
>
> For different **sensor modality & detection model** combinations, we encourage reviewers to refer to Table 3, where we have showcased five distinct sensor setups and detection model combinations. This comprehensive analysis provides valuable insights into how our methodology adapts to and performs with various sensor configurations and detection algorithms, highlighting its versatility and effectiveness in diverse operational contexts.
>
> >**W5:** The 5 hour training time is for single modality. What is the overall training time and the associated time for the compared model?
>
> The overall training time is 8 hours.
>
> One of the significant advantages of HEAL is that new agent type be trained in parallel, which does not require collaboration. So the overall training time is 5 (base training) + 3 (new agent type training) = 8 hours. Even with the continuous increase in agent types, we can maintain the training duration within 8 hours. This efficiency in training time underscores the scalability and effectiveness of our training strategy.
>
> In comparison, the training time is notably huge for other compared models, such as V2X-ViT and HM-ViT. As the number of agent types increases, the training time will continue to grow because more encoder parameters will be updated. They also require more epochs to fit complex transformer structures. In our experiments, 50 epochs are set to ensure convergence (validation loss is minimized). Under the combination of 4 agent types, with the same training range of x in [-102.4m, -102.4m] and y in [-51.2m, 51.2m], the transformer-based model training takes about 30 hours using 2 RTX 3090. Since HM-ViT constructs pairwise attention weight learning for 4 heterogeneous agent types and iterates each agent as ego, the complexity is further increased, taking about 40 hours.

---

> ### Author Response · Authors · 2023-11-19
> **Response #5**
>
> >**Q1:** The author argued that "late fusion is suboptimal due to the communication latency". However, as shown in DiscoNet/V2X-ViT/V2VNet etc., intermediate fusion methods usually require larger bandwidth requirements than late fusion, which can even lead to potentially larger communication latency compared with late fusion.
>
>
> The communication cost of intermediate fusion can be significantly reduced with technical designs.
>
> 1. As shown in V2X-ViT/V2VNet/OPV2V **and our paper (Figure 4)**, we can compress the intermediate feature to a small data size *along channel dimension* with an autoencoder structure. In Figure 4 of our paper, we present experiments that compress the channel number of features for communication, and it performs well.
>
> 2. As shown in Where2comm, we can also compress the feature *along spatial dimension*. We can select the most important feature (selected by our foreground estimator) for transmission. In Where2Comm's Appendix Table 3 and Figure 12, intermediate fusion with this technique has a communication volume already comparable to late fusion.
>
> Therefore, the communication cost problem can be solved in many ways, and it is also shown in our paper in Figure 4. Actually, there are currently many papers focusing on communication bandwidth and delay. These issues are significant but are not the focus of this work.
>
> >**Q2:** How the result of HM-ViT is reproduced? Is the heterogeneity used for all 4 modalities? Or only two modalities are used as the original paper?
>
> We carefully reproduce this model ourselves because the authors have not released the code. To replicate HM-ViT, we adhered strictly to the model design and algorithmic procedures outlined in the original paper. We meticulously followed the process described in Algorithm 1 from the paper. This included updating the features of different agents while iterating each as an ego, alternately using Local and Global hetero-attention blocks, and configuring distinct MLP and LN layers for different agent types. We set the same window size and feature channel as in the original study to ensure the accuracy of our replication. The baseline performance comparisons in our Appendix align with those in the HM-ViT paper, validating the effectiveness of our replication effort.
>
> The heterogeneity is used for 4 modalities. The HM-MLP, HM-LN, HM-decoder and H$^3$GAT are all set with 4 agent types.
>
> We express our profound gratitude for the exceptional contributions made by HM-ViT to the community, which have significantly inspired our contemplation on heterogeneous collaborative perception, culminating in the creation of this present article.
>
> >**Q3:** What is the inference time of the proposed method?
>
> The table below shows the inference time of all methods on one RTX A40 with batchsize 1. We see that HEAL is running at an acceptable FPS.
>
> | Inference FPS (fusion module)  on OPV2V-H dataset | - | - | - | - | - |
> |---------------------------------------------------|--|--|--|--|--|
> | range $x \in$  [-102.4m, 102.4m], $y \in$ [-51.2m, 51.2m] | $N_{agent}$= 1 | $N_{agent}$= 2 | $N_{agent}$= 3 | $N_{agent}$= 4 | $N_{agent}$= 5 |
> | AttFusion | 500.3 | 359.9 | 248.0 | 207.6 | 159.1 |
> | DiscoNet | 477.6 | 383.3 | 292.6 | 235.8 | 196.3 |
> | F-Cooper | 1025.7 | 869.4 | 757.3 | 681.0 | 595.5 |
> | V2XViT | 54.1 | 30.83 | 19.78 | 15.50 | 12.07 |
> | CoBEVT | 112.4 | 67.16 | 45.57 | 33.78 | 24.89 |
> | HM-ViT | 6.08 | 5.56 | 4.60 | 3.95 | 2.99 |
> | HEAL | 55.8 | 48.07 | 42.35 | 35.21 | 30.04 |

---

> ### Author Response · Authors · 2023-11-19
> **Response #6**
>
> >**Q4:** What is the influence of the modality choice in the base collaboration training?
>
> We really appreciate insightful thoughts and the highly meaningful question. Choosing agents with LiDAR modality as the collaboration base yields better results than using agents with camera modality. It is intuitive that LiDAR, compared to cameras, often achieves higher detection accuracy, which is beneficial for feature learning and helps construct a robust unified feature space with clear foreground-background differentiation. In our agent configuration, even though we provide depth information as supervision for cameras, their detection performance still falls short compared to LiDAR. Consequently, a HEAL model trained with a camera collaboration base exhibits a noticeable decrease in performance. In this ablation study, the agent combination is still  $\mathbf{L}_p^{(64)}$ +$\mathbf{C}^{(384)}_E$ +$\mathbf{L}^{(32)}_S$ +$\mathbf{C}^{(336)}_R$, but we change use different base agent type as the base type. The results are as follows. We see that using LiDAR as the collaboration base is significantly better than using cameras.
>
>
> | **Base Agent Type Selection** | **AP30** | **AP50** | **AP70** |
> |-------------------------------|----------|----------|----------|
> | $\mathbf{L}_p^{(64)}$ | 0.908 | 0.894 | 0.813 |
> | $\mathbf{L}^{(32)}_S$ | 0.904 | 0.892 | 0.805 |
> | $\mathbf{C}^{(384)}_E$ | 0.804 | 0.794 | 0.714 |
>
>
> **In real world, employing LiDAR modality to train a collaboration base is a totally viable and worthwhile solution.** This is also supported by existing datasets such as DAIR-V2X and V2V4Real, which provide the necessary data for such training. We have detailed our approach for practical implementation in the Appendix of our paper, emphasizing its feasibility and relevance in real-world applications.

---

> ### Comment · Reviewer_bqtt · 2023-11-21
>
> For W1 (b), I appreciate the author for highlighting then novelty of the training strategy. However, still, it is not clear if the performance boost is mainly from the training strategy or the fusion model. I would encourage the author to compare different fusion models (these can use the proposed training strategy) with and without the new training strategy. In this way, the true performance boost brought by the proposed new fusion method can be demonstrated.

---

> > ### Author Response · Authors · 2023-11-22
> >
> > > For W1 (b), I appreciate the author for highlighting then novelty of the training strategy. However, still, it is not clear if the performance boost is mainly from the training strategy or the fusion model. I would encourage the author to compare different fusion models (these can use the proposed training strategy) with and without the new training strategy. In this way, the true performance boost brought by the proposed new fusion method can be demonstrated.
> >
> > Thank you for your suggestion. In our previous response, we have shown (i) the performance of all methods trained collectively, as well as our HEAL with novel backward alignment, (ii) the performance of different fusion modules under pure LiDAR collaboration and pure camera collaboration. This illustrates the superiority of pyramid fusion and that **major performance improvements in open heterogeneous collaborative perception scenarios come from backward alignment.**
> >
> > We also elaborate on why not all fusion modules are compatible with backward alignment, and our Pyramid Fusion is a reasonable and superior design in this context. But your suggestions are constructive; we are adding relevant experiments, and we appreciate your patience.

---

> ### Author Response · Authors · 2023-11-22
>
> > For W1, (a) the Node level weight prediction: it is still due to the proposed training strategy not the novel architecture/model-level design. (b) Explicit foreground estimators: Why Where2comm etc. lack an estimator and supervision? Where2comm uses the detection head (generator in equation (1) in the original where2comm paper) to generate the spatial confidence map. This detection head is trained and supervised with the training data as it is also used for generating the bounding box proposals. Please highlight the difference.
>
>
>
> (a) The process of estimating node-level weights **is clearly a part of the model design** because it generate foreground score maps (weight) from single-agent features.
>
> (b) There are three significant differences between the detection head (Where2comm) and the foreground estimator (HEAL).
>
> 1. The detection head (Where2comm) aims to filter critical features for transmission; while the foreground estimator (HEAL) is used to enhance feature fusion. In where2comm, the detection head is used to generate a spatial confidence map, which is used to filter features for bandwidth-saving transmission. Its behavior is a 01 mask, which only sparses the sent feature map and **does not participate in any subsequent feature fusion**. Subsequent feature fusion is achieved through a Transformer structure. In HEAL, the foreground score estimated by the foreground estimator will directly participate in feature fusion and be used to adjust the weight of each agent at each BEV pixel position during fusion, allowing the HEAL to **directly encode the different importances of different agents into the final fused feature**.
> 2. In Where2comm, initial detection head and final detection head share the same structures and parameters; while in heal, foreground estimator and the final detetion head are decoupled.  The single-agent detection head and the final detection head in Where2comm share a set of parameters, which significantly constrains the learning of features. In contrast, the HEAL framework's foreground estimator and the final detection head are separate, operating with two distinct network structures and sets of parameters. **This decoupling allows our Pyramid structure to fully explore and learn from features across different scales.** More importantly, the use of an independent foreground estimator enhances the effectiveness of feature alignment in new agents, as it focuses solely on the characteristics of individual features, aligning with our targeted objectives.
> 3. Since detection head (Where2comm) is a copy of the final classification head, it is trained without the individual supervision, while HEAL devises an explicit supervision on the foreground score map. The foreground supervision employed by our HEAL plays two essential roles: First, it enhances the training  of foreground estimators, resulting in more precise predictions of  weights. Second, and more importantly, it provides supervision signals for feature alignment during the backward alignment process.  As the parameters of the foreground estimator remain fixed when training new heterogeneous agents, they undergo the same individual supervision for foreground estimation. This process motivates the encoder to generate features that are in alignment with those of the base agent.

---

> ### Comment · Reviewer_bqtt · 2023-11-22
>
> The arguments are contradictory. In the original paper, the author argued "late fusion is suboptimal due to the communication latency". However, in the rebuttal phase, the author argued though intermediate fusion may have larger bandwidth requirements but still "compressing feature channel can reduce communication latency to a fully acceptable value". Then what is the advantages of using intermediate fusion in terms of latency. Certain methods like V2X-ViT have specific design to deal with latency. However this work doesn't have specific module for time delay. Moreover, in most cooperative perception papers, it is acknowledged that late fusion has far less bandwidth requirements than other fusion strategies. Thus I don't think it is fair to argue late fusion is suboptimal due to communication delay, which is used to motivate using intermediate fusion.
>
> What is the data transmission size for the proposed method? Would the pyramid structure require multiple rounds of communications?

---

> ### Comment · Reviewer_bqtt · 2023-11-22
>
> For W1(b), the author only compared with where2comm and who2comm. Comparing with more SOTA methods are required to fully justify the performance boost brought by the proposed pyramid fusion component. I understand due to the time limit, demonstrating the results in the rebuttal phase may not be practical. Still, I think the final version would benefit from these experiments.

---

> ### Author Response · Authors · 2023-11-22
>
> > (a) The arguments are contradictory. In the original paper, the author argued "late fusion is suboptimal due to the communication latency". However, in the rebuttal phase, the author argued though intermediate fusion may have larger bandwidth requirements but still "compressing feature channel can reduce communication latency to a fully acceptable value". (b) What is the data transmission size for the proposed method?   (c) Would the pyramid structure require multiple rounds of communications?
>
> (a) We are sorry about any confusion, but we do not understand what the contradiction the reviewer mentioned is. We never claimed "late fusion is suboptimal due to the communication latency" and " it is advantageous of using intermediate fusion in terms of latency". **Please do not misrepresent our paper to create controversy.**
>
> The motivation for late fusion is derived from Page 2 of our paper. It stated that "However, its (late fusion) performance is suboptimal and has been shown particularly vulnerable to localization noise and communication latency" in our original paper. This sentence claims two drawbacks of late fusion: (i) when the communication environment is ideal, the detection performance of late fusion is worse than intermediate fusion (ii) when considering the effect of pose error and latency, late fusion is more vulnerable than intermediate fusion. The reason for choosing intermediate fusion is precisely because of these two shortcomings of late fusion:
>
> 1. Suboptimal detection performance.
>
>    The late fusion approach, as evidenced by key studies in collaborative perception [1,2,3,4,5,6,7,8,9,10], are less effective than intermediate fusion methods. As shown in HM-ViT [6], in terms of AP70, the intermediate fusion performance can exceed the late fusion by 40.2%. This is the main motivation for using intermediate fusion.
>
> 2. Vulnerability to pose noise and communication latency.
>
>    Late fusion fuses the bounding boxes from each agent, but this process is sensitive to disturbances like noise in pose or communication delays.  Such interferences can cause spatial misalignment of the transmitted bounding box, significantly degrading performance [8,9]. In contrast, using intermediate fusion can significantly enhance model's robustness to these interferences, because the features have a larger receptive field and thus are not directly fatally affected like bounding boxes when interferences exist.
>
> (b) Regarding data transmission size, it is all the same across the intermediate fusion methods in our experiments. In the original paper, the data size without any compression is 32MB (AP70=0.813); after 32x compression, the data size is 1MB (AP70=0.806); after spatial filtering (HEAL selects the features according to the foreground score), the transmitted data size is 0.004MB (AP70=0.801).
>
> (c) There is **only one round** of communication. Compared with V2X-ViT and AttFuse, which require two rounds of communication (one for transmitting poses and one for transmitting features), our design is more rational.
>
> | Methods | Communication Round $\downarrow$ | Channel Compress | Spatial Compress |
> |---------|----------------------------------|------------------|------------------|
> | AttFuse[1] | 2 | $\checkmark$ | $\times$ |
> | V2X-ViT[2] | 2 | $\checkmark$ | $\times$ |
> | **HEAL** | 1 | $\checkmark$ | $\checkmark$ |
>
> As we have repeatedly emphasized, **our focus is not on communication compression or latency compensation in collaborative perception**. Instead, we focus on integrating new agents into the collaborative framework with high performance and low training costs. We hope that the reviewer can recognize the aim of this paper and its practical implications, including: (i) reducing the multi-agent data collection costs for new heterogenous agents, (ii) reducing the training cost while maintaining high performance; and (iii) addressing the model detail privacy concerns.
>
> *References*
>
> [1] OpV2V: An open benchmark dataset and fusion pipeline for perception with vehicle-to-vehicle communication, ICRA 2022.
>
> [2] V2X-ViT: Vehicle-to-everything cooperative perception with vision transformer. ECCV 2022.
>
> [3] V2v4real: A real-world large-scale dataset for vehicle-to-vehicle cooperative perception. CVPR 2023
>
> [4] V2xp-asg: Generating adversarial scenes for vehicle-to-everything perception. ICRA 2023
>
> [5] Collaboration Helps Camera Overtake LiDAR in 3D Detection. CVPR 2023
>
> [6] HM-ViT: Hetero-modal Vehicle-to-Vehicle Cooperative perception with vision transformer, ICCV 2023
>
> [7] CoBEVT: Cooperative bird's eye view semantic segmentation with sparse transformers. CoRL 2022.
>
> [8] Robust Collaborative 3D Object Detection in Presence of Pose Errors. ICRA 2023
>
> [9] Asynchrony-Robust Collaborative Perception via Bird’s Eye View Flow. NeurIPS 2023.
>
> [10] Where2comm: Communication-Efficient Collaborative Perception via Spatial Confidence Maps. NeurIPS 2022

---

> ### Author Response · Authors · 2023-11-22
>
> > For W1(b), the author only compared with where2comm and who2comm. Compared with more SOTA methods are required to fully justify the  performance boost brought by the proposed pyramid fusion component.
>
> We have provided the benchmark under the same collective training strategy with 3 settings:
> - (i) pure LiDAR-based collaboration
> - (ii) pure camera-based collaboration
> - (iii) open heterogeneous collaborative perception
>
> For (i) pure LiDAR-based collaboration & (ii) pure camera-based collaboration, the table below reflects that 1) Pyramid Fusion has significant detection performance improvements compared to previous methods, especially on AP70. 2) On the real-world data set DAIR-V2X, this conclusion still holds, and our Pyramid Fusion can show better performance. They are all compared very fairly, with only the difference being the fusion module.
>
> | Dataset | OPV2V | - | - | - | DAIR-V2X | - | - | - |
> |---------|-------|---|---|---|----------|---|---|---|
> | **Method** | LiDAR-based | LiDAR-based | Camera-based | Camera-based | LiDAR-based | LiDAR-based | Camera-based | Camera-based |
> | - | AP50 | AP70 | AP50 | AP70 | AP30 | AP50 | AP30 | AP50 |
> | No Fusion | 0.782 | 0.634 | 0.405 | 0.216 | 0.421 | 0.405 | 0.014 | 0.004 |
> | F-Cooper | 0.821 | 0.632 | 0.469 | 0.219 | 0.723 | 0.620 | 0.115 | 0.026 |
> | DiscoNet | 0.882 | 0.737 | 0.517 | 0.234 | 0.746 | 0.685 | 0.083 | 0.017 |
> | AttFusion | 0.878 | 0.751 | 0.529 | 0.252 | 0.738 | 0.673 | 0.094 | 0.021 |
> | V2XViT | 0.917 | 0.790 | 0.603 | 0.289 | 0.785 | 0.521 | 0.198 | 0.057 |
> | CoBEVT | 0.935 | 0.821 | 0.571 | 0.261 | 0.787 | 0.692 | 0.182 | 0.042 |
> | HMViT | 0.950 | 0.873 | 0.643 | 0.370 | 0.818 | 0.761 | 0.163 | 0.044 |
> | **Pyramid Fusion** | **0.963** | **0.926** | **0.689** | **0.468** | **0.832** | **0.790** | **0.282** | **0.128** |
>
> For (iii) open heterogeneous collaboration results,  we compare various fusion methods based on the same collective training strategy. The table below reflects that: 1) Pyramid Fusion still maintains relatively leading performance; 2) for every new heterogeneous agent added, Pyramid Fusion can still have better detection accuracy.
>
> | OPV2V-H | AP70 | AP70 | AP70 |
> |--------------------------|------|--|--|
> | **Method** | $\mathbf{L}_p^{(64)}$ +$\mathbf{C}^{(384)}_E$ | $\mathbf{L}_p^{(64)}$ +$\mathbf{C}^{(384)}_E$ +$\mathbf{L}^{(32)}_S$ | $\mathbf{L}_p^{(64)}$ +$\mathbf{C}^{(384)}_E$ +$\mathbf{L}^{(32)}_S$ +$\mathbf{C}^{(336)}_R$ |
> | No Fusion | 0.606 | 0.606 | 0.606 |
> | Late Fusion | 0.599 | 0.685 | 0.685 |
> | FCooper | 0.628 | 0.517 | 0.494 |
> | DiscoNet | 0.653 | 0.682 | 0.695 |
> | AttFuse | 0.635 | 0.685 | 0.659 |
> | V2XViT | 0.655 | 0.765 | 0.753 |
> | CoBEVT | 0.671 | 0.742 | 0.742 |
> | HMViT | 0.646 | 0.743 | 0.755 |
> | **Pyramid Fusion** | **0.690** | **0.768** | **0.769** |
>
> In summary, all the results have validated that **with the same collective training strategy, Pyramid Fusion outperforms all the baseline fusion methods.** Note that to effectively compare the performance of feature fusion modules, the most advantageous approach is to assess all fusion module with the collective training strategy, as (i) (ii) and (iii) do. This recommendation stems from our explicit clarification that not all fusion methods are compatible with backward alignment, a conclusion substantiated by experimental evidence (in Response #2).
>
> > I would encourage the author to compare different fusion models (these can use the proposed training strategy) with and without the new training strategy.
>
> According to the reviewer's original comment, we have finished all the experiments for comparing different fusion models without the new training strategy and the results have been shown above.
>
> Now, to address the reviewer's latest demand, we are trying our best to conduct new experiments for comparing different fusion models with the new training strategy. Please be patient.

---

> ### Author Response · Authors · 2023-11-23
>
> We thank the reviewers for their patience and we show the results of training V2X-ViT and HM-ViT using our backward alignment strategy. The results are as follows.
>
> | AP70                                              | $\mathbf{L}_p^{(64)}$ +$\mathbf{C}^{(384)}_E$ | $\mathbf{L}_p^{(64)}$ +$\mathbf{C}^{(384)}_E$ +$\mathbf{L}^{(32)}_S$ | $\mathbf{L}_p^{(64)}$ +$\mathbf{C}^{(384)}_E$ +$\mathbf{L}^{(32)}_S$ +$\mathbf{C}^{(336)}_R$ |
> | ------------------------------------------------- | --------------------------------------------- | ------------------------------------------------------------ | ------------------------------------------------------------ |
> | V2X-ViT + Collective Training                     | 0.655                                         | 0.765                                                        | 0.753                                                        |
> | HM-ViT + Collective Training                      | 0.646                                         | 0.743                                                        | 0.755                                                        |
> | Pyramid Fusion + Collective Training              | 0.690                                         | 0.768                                                        | 0.769                                                        |
> |                                                   |                                               |                                                              |                                                              |
> | V2X-ViT + Back. Align. Training                   | 0.664                                         | 0.677                                                        | 0.675                                                        |
> | HM-ViT + Back. Align. Training                    | 0.630                                         | 0.622                                                        | 0.615                                                        |
> | **Pyramid Fusion + Back. Align. Training (HEAL)** | **0.726**                                     | **0.812**                                                    | **0.813**                                                    |
>
> We see that (i) Under the same collective training strategy, Pyramid Fusion surpasses existing methods like V2X-ViT and HM-ViT. (ii) Under the backward alignment based training, HEAL demonstrates significant performance improvements. This can be attributed to the fact that backward alignment mitigates the issues of imbalance and inadequacy in collective training when faced with a high number of agent types. In contrast, when trained using backward alignment, V2X-ViT and HM-ViT's performance is even inferior to collective training. This performance degradation indicates that not all fusion modules are compatible with backward alignment, particularly those that heavily rely on edge interaction between agents. We will elaborate on it in the next paragraph
>
> Fusion methods that rely on edge interaction are difficult to fully benefit from the backward alignment training method. In the case of HM-ViT with four agent types present in the scene, a collaboration graph is constructed with four vertices representing each agent type. However, under the backward alignment training strategy, the new agent is only trained as a single agent without collaboration, so the heterogeneous edges between different agent types are never updated. This significantly limits the performance in the inference stage and makes the collaboration weights erroneously estimated between agents of different types.
>
> Therefore, we clarify that our Pyramid Fusion and backward alignment complement each other. Pyramid Fusion + backward alignment makes HEAL optimal in terms of both collaboration performance and training costs.
>
> Lastly, we kindly request the reviewer's attention to these real-world implications of HEAL, as they highlight its practical advantages in terms of data collection cost reduction, training cost reduction, and model-detail privacy preservation. It is our collective aspiration to enable the deployment of collaborative perception on real-world vehicles, making traffic more efficient and enhancing the safety on the road.
>
> Sincerely.

---

### Official Review · Reviewer_wcWo · 2023-10-29

**Soundness:** 3 good
**Presentation:** 3 good
**Contribution:** 3 good
**Rating:** 6
**Confidence:** 5

**Summary:**

This paper introduces HEterogeneous ALliance (HEAL), an innovative framework designed to enhance collaborative perception by accommodating continually emerging heterogeneous agent types, addressing the domain gap issue in existing systems. The framework establishes a unified feature space that aligns new, diverse agents through a cost-effective and secure backward alignment process, requiring only individual training for the new agents. This approach not only minimizes training costs and maximizes extensibility but also safeguards the model details of the new agents. The authors also introduce a new extensive dataset, OPV2V-H, to advance research in heterogeneous collaborative perception. Experiments reveal that HEAL outperforms state-of-the-art methods, showing a remarkable reduction in training parameters by 91.5% while integrating three new agent types.

**Strengths:**

1. The paper introduces a interesting open heterogeneous collaborative perception setting. Agents with different sensor can collaborate for vision tasks. This is an interesting and practical setting.
2. Multi-scale feature fusion and the 'late participation' strategy is reasonable for such tasks.
3. A dataset contribution. Experiments are extensive. Presentation of the paper is good.

**Weaknesses:**

1. There is no real-world experiments. There are some dataset like nuScene/nuPlan, Waymo and etc including data of different sensors. It would be nice to show some real examples.
2. It would be interesting to include a bit discussion on related works for cooperation for driving tasks, e.g. [1][2][3]
3. I don't find a code release. Would be nice to release the code for supplementary or public github repo.

[1] D Chen and et al. Learning from All Vehicles. CVPR 2022.
[2] J Cui and et al. Coopernaut: End-to-end driving with cooperative perception for networked vehicles. CVPR 2022.
[3] R Zhu and et al. Learning to Drive Anywhere. CoRL 2023.

**Questions:**

1. The paper mentioned new agent privacy issue. I assume the late participate will require the new agent to access the fused feature to do the update according to equations in Section 4.3. Will the fused feature release some privacy of the old agents to the new agent?
2. For the experiments setting, are there any case agent exits the cooperation during training? How the framework will do to deal with this situation.
3. I am wondering if there are some simple experiments to show real world cases as I mentioned in the weak point.

---

> ### Author Response · Authors · 2023-11-19
> **Response #1**
>
> To the esteemed reviewer, I deeply appreciate the time and effort you've put into reviewing my work. Your suggestions are very helpful and will be carefully considered. Now let me address some concerns for you.
>
> > **W1:** There is no real-world experiments. There are some dataset like nuScene/nuPlan, Waymo and etc including data of different sensors. It would be nice to show some real examples.
>
> The DAIR-V2X[4] dataset used in our paper is a **real-world** multi-agent collaborative perception dataset. It is collaboratively collected by a vehicle and a roadside unit in the real world, equipped with multiple sensors, including cameras and LiDAR. In our Section 5.1, we provide a brief introduction to the dataset, and Section 5.3 is dedicated to extensive experimentation on it.
>
> nuScene, nuPlan and Waymo are all awesome autonoumous driving datasets, but they has only one data collection vehicle. They do not have temporally and spatially aligned multi-vehicle data, making them difficult to validate on our task. We believe that DAIR-V2X, as a real dataset widely used in collaborative perception (e.g. [6][7][8][9][10]), can verify the reliability of our algorithm.
>
> In our paper's Table 3, we conducted experiments using the real-world DAIR-V2X dataset under our open heterogeneous collaborative perception setting. We tested three different combinations of agent types, and the results are presented as follows:
>
>
> | Agent Types | $\mathbf{L}_p^{(40)}$ +$\mathbf{C}^{(288)}_E$ | - | $\mathbf{L}_p^{(40)}$+$\mathbf{L}^{(40)}_S$ | - | $\mathbf{L}_p^{(40)}$ +$\mathbf{C}^{(288)}_R$ | - |
> |-------------|-----------------------------------------------|--|---------------------------------------------|--|-----------------------------------------------|--|
> | - | AP50 $\uparrow$ | Training Param (M) $\downarrow$ | AP50 $\uparrow$ | Training Param (M) $\downarrow$ | AP50 $\uparrow$ | Training Param (M) $\downarrow$ |
> | Late Fusion | 0.344 | 25.7 | 0.376 | 11.8 | 0.341 | 12.6 |
> | F-Cooper | 0.626 | 38.8 | 0.545 | 24.9 | 0.611 | 25.8 |
> | DiscoNet | 0.576 | 38.9 | 0.634 | 25.1 | 0.621 | 25.9 |
> | AttFusion | 0.649 | 38.8 | 0.661 | 24.7 | 0.623 | 25.8 |
> | V2XViT | 0.654 | 49.6 | 0.709 | 35.7 | 0.669 | 36.6 |
> | CoBEVT | 0.638 | 43.8 | 0.692 | 29.8 | 0.66 | 30.7 |
> | HM-ViT | 0.638 | 67.8 | 0.538 | 53.9 | 0.677 | 54.8 |
> | **HEAL** | **0.658** | **25.7** | **0.770** | **11.8** | **0.681** | **12.6** |
>
>
>
> The experimental outcomes align with those demonstrated on the OPV2V-H: HEAL outperforms other methods while maintaining the lowest training cost, validating the high performance and low training costs of our methods.
>
> > **W2:** It would be interesting to include a bit discussion on related works for cooperation for driving tasks, e.g. [1][2][3]
>
> Thank you very much for recommending these driving related articles. In the revision, We have cited [1,2,3] and add discussions in our related works. We believe this will enhance the depth of the paper and inspire the readers.
>
>
> Here, we provide a detailed discussion about how collaborative perception relates to cooperation in driving tasks in two main ways:
> 1) [1] and [3] concentrate on enhancing model training by incorporating multiple sources beyond just the ego agent. Collaborative perception benefits these agents by providing additional, complementary information during inference. This collaborative approach in driving tasks offers significant insights for improving the training process, thereby enhancing the generalizability of collaborative perception. Furthermore, collaborative perception can improve cooperative driving tasks by integrating collaboration modules into single-agent models, enhancing performance during inference.
> 2) [2] explores how sharing information benefits the ego agent's end-to-end driving task, which closely aligns with our collaborative perception approach. Investigating collaborative perception can further promote cooperation in driving tasks, as a more holistic perception contributes to overall driving enhancements. For example, our research, aimed at expanding the scope of collaboration, can be applied to collaborative driving systems to improve their overall efficiency and effectiveness.

---

> ### Author Response · Authors · 2023-11-19
> **Response #2**
>
> > **W3:** I don't find a code release. Would be nice to release the code for supplementary or public github repo.
>
> We plan to release the code if the paper can be accepted. In the meantime, we are happy to discuss any technical detail to address your concerns and support reproducibility. We appreciate your understanding and look forward to sharing our complete code soon.
>
> > **Q1:** The paper mentioned new agent privacy issue. I assume the late participate will require the new agent to access the fused feature to do the update according to equations in Section 4.3. Will the fused feature release some privacy of the old agents to the new agent?
>
> Thanks for your thoughtful question! There is no clear privacy leakage from the old agents to the new agents in inference stage. The privacy concerns discussed in our paper primarily relate to the model-detail privacy, which arises from collective training. During collective training, each agent type must provide its network structure for gradient backpropagation and parameter updates.
>
> In response to this question, we provide insights from two perspectives: sensor-data privacy and model-detail privacy.
>
> - **Sensor-data privacy:** During the inference process, the transmitted features are derived from observations of the current scene. What they describe is a public driving scenario, so there are no privacy issues at the sensor data level.
>
> - **Model-detail privacy:** In the inference stage, it is challenging to deduce the model details of different agent types based solely on features, especially when both the network structure and parameters are unknown. In HEAL, the collaboration network (excluding the encoder) trained by base agents will be distributed to other agents, which may cause some concerns about leakage of the base agent model details. But a highly feasible solution is to use public multi-agent collaborative perception datasets for the collaboration base training, such as DAIR-V2X[4] and V2V4real[5]. This will completely eliminate model detail privacy issues in HEAL. We have delineated such a real-world solution in the Appendix, which comprehensively prevent the exposure of model privacy.
>
> >**Q2:** For the experiments setting, are there any case agent exits the cooperation during training? How the framework will do to deal with this situation.
>
> The number of input agents will not affect the collaboration process. This is equivalent to the situation you mentioned when an agent quits cooperation.
>
> The number of agents in the OPV2V-H data set varies from 2-7 and is not a fixed value itself. Therefore, our network is already trained with varying agent numbers. We use the foreground estimator for each agent to estimate its score map, normalize them with softmax, and then aggregate them into a weighted sum. For any number of agents, the aggregated features can always be considered to be the most important foreground features from all participants.
>
> > **Q3:** I am wondering if there are some simple experiments to show real world cases as I mentioned in the weak point.
>
> We conduct extensive experiment on DAIR-V2X[4] dataset, which is a large real-world multi-agent perception dataset. The experimental results on real-world data validate the effectiveness of our approach. See more details in response to **W1**.
>
>
> [1] Learning from All Vehicles. CVPR 2022.
>
> [2] Coopernaut: End-to-end driving with cooperative perception for networked vehicles. CVPR 2022.
>
> [3] Learning to Drive Anywhere. CoRL 2023.
>
> [4] DAIR-V2X: A Large-Scale Dataset for Vehicle-Infrastructure Cooperative 3D Object Detection. CVPR 2022.
>
> [5] V2V4Real: A large-scale real-world dataset for Vehicle-to-Vehicle Cooperative Perception. CVPR 2023.
>
> [6] Asynchrony-Robust Collaborative Perception via Bird’s Eye View Flow. NeurIPS 2023.
>
> [7] Robust Collaborative 3D Object Detection in Presence of Pose Errors. ICRA 2023
>
> [8] Where2comm: Communication-Efficient Collaborative Perception via Spatial Confidence Maps. NeurIPS 2022
>
> [9] TransIFF: An Instance-Level Feature Fusion Framework for Vehicle-Infrastructure Cooperative 3D Detection with Transformers. ICCV 2023.
>
> [10] Collaboration Helps Camera Overtake LiDAR in 3D Detection. CVPR 2023.

---

> ### Author Response · Authors · 2023-11-22
>
> We are grateful once again for your invaluable review. In our response, we have endeavored to thoroughly address all your concerns and questions:
>
> - We elucidate the experiments conducted on the real-world dataset DAIR-V2X[1], affirming the efficacy of our approach in reducing the cost of training while maintaining high collaborative performance in real-world scenarios.
> - We also expound that HEAL remains effective across different agent numbers in collaboration (just like if an agent quits the collaboration) because the Pyramid Fusion always selects the most critical features to fuse.
> - Furthermore, we articulate that there is no model detail disclosure within the framework of HEAL, even from the base agents. This is substantiated in our Appendix, where we comprehensively expose our practical solutions to this concern.
>
> We really appreciate your valuable time to review our updated response and verify if it satisfactorily answers your questions. We are eager to engage in further discussion and offer additional clarifications for any new inquiries you may have.
>
> References:
>
> [1] DAIR-V2X: A Large-Scale Dataset for Vehicle-Infrastructure Cooperative 3D Object Detection. CVPR 2022.

---

### Official Review · Reviewer_hUnZ · 2023-11-02

**Soundness:** 3 good
**Presentation:** 3 good
**Contribution:** 3 good
**Rating:** 8
**Confidence:** 5

**Summary:**

This paper presents a novel question surrounding the task of multi-agent collaborative perception, duly considering the scenario where new collaborators continually join the perception system in real-world settings. This is a highly intriguing question, directly impacting the deployment of multi-agent collaborative perception systems.

The author proposes a highly concise solution, introducing the HEAL framework, which is capable of accommodating the features acquired by the agents newly joining the system. The experimental results demonstrate that the proposed new framework is effective in accommodating agents newly incorporated into the system, and achieves state-of-the-art results.

In conclusion, this is a highly intriguing piece of work.

**Strengths:**

1. Presents a highly intriguing new question, effectively addressing the challenges faced during the deployment of multi-agent collaborative perception systems.

2. The approach in this paper is notably succinct and efficient; the authors design a novel backward alignment mechanism for individual training. This method constructs an alignable feature space, facilitating subsequent updates of features transmitted by other agents.

**Weaknesses:**

1. The intermediate fusion method employed in this paper doesn't seem to address the issue of new agents joining as effectively as late fusion does.

2. This paper has only conducted experiments on two datasets, one of which is generated for the first time in this paper. It is hoped that the author can introduce more experiments to substantiate.

**Questions:**

1. In this paper, the authors claim that agents newly joining the system may struggle to align well in the feature space due to data distribution differences. However, is training a unified feature space an effective solution? Given that data discrepancies arising from different sensors inherently result in domain differences, this discrepancy poses a significant challenge in the domain adaptation field. Is the method proposed in this paper suitable for addressing this issue?

2. The author raises a novel question, thus it would be prudent to utilize more datasets to verify the efficacy of the proposed method. This is because some schemes[1] solely employing distillation can achieve significant improvements in accuracy. In the open-source datasets they used, there are also newly joining agents, similar to the DAIR-V2X dataset used in this paper. It is hoped that the author can supplement with more extensive experiments to substantiate the reliability of the raised question and the effectiveness of the proposed method.

3. Referring to the article on late fusion[2], I believe that late fusion seems to be a more effective solution to the problem posed in this paper. While the process of late fusion indeed has some issues with error accumulation, [2] has adeptly mitigated some of the past problems of late fusion through trajectory prediction. At the same time, employing late fusion can maximally avoid the issue of aligning features extracted by different agents, fundamentally resolving the problem posed in this paper. I hope that the author can conduct a comparative analysis between the methods of these two papers.

[1]. Z. LI, et al. MKD-Cooper: Cooperative 3D Object Detection for Autonomous Driving via Multi-teacher Knowledge Distillation. IEEE Transactions on Intelligent Vehicles, 2023.

[2]. S. Wei, et al. Asynchrony-Robust Collaborative Perception via Bird's Eye View Flow. NIPS 2023.

---

> ### Author Response · Authors · 2023-11-19
> **Response #1**
>
> Dear Reviewer,
>
> First and foremost, I would like to extend my sincere gratitude for your time and effort in reviewing the submission. Your insightful comments and constructive suggestions are invaluable to refining this work. In the following, we will address your concerns regarding the weaknesses and questions and show additional experimental results.
>
> > **W1:** The intermediate fusion method employed in this paper doesn't seem to address the issue of new agents joining as effectively as late fusion does.
>
> HEAL achieves significantly better detection performance than late fusion. As shown in Table 3 from our paper:
>
> | AP70 | OPV2V-H | OPV2V-H | OPV2V-H |
> |---|--|---|---|
> | **Agent Types** | $\mathbf{L}_p^{(64)}$ +$\mathbf{C}^{(384)}_E$ +$\mathbf{L}^{(64)}_S$ + $\mathbf{C}^{(336)}_R$ | $\mathbf{L}_p^{(32)}$ +$\mathbf{C}^{(384)}_E$ +$\mathbf{L}^{(32)}_S$ + $\mathbf{C}^{(336)}_R$ | $\mathbf{L}_p^{(16)}$ +$\mathbf{C}^{(384)}_E$ +$\mathbf{L}^{(16)}_S$ + $\mathbf{C}^{(336)}_R$ |
> | Late Fusion | 0.685 | 0.639 | 0.457 |
> | HEAL | 0.813 | 0.738 | 0.578 |
>
> we see that HEAL significantly outperforms the late fusion with our feature-level fusion.
>
> Furthermore, HEAL is as efficient as late fusion regarding new agent integration in training.  The new agent type's training process in HEAL is also performed on single agents without collaboration. This enables us to maintain comparable training parameters, FLOPs, and training time with late fusion.
>
> > **W2:** It is hoped that the author can introduce more experiments to substantiate.
>
> We add experiments on V2XSet[1] and OPV2V(culver city)[2]. We prepare the similar agent type setting in OPV2V-H and evaluate the collaborative scene-level AP when gradually adding new heterogeneous agents into the scene. The experimental results are as follows:
>
> *  V2XSet dataset. We see that HEAL consistently achieves the best performance each time new heterogeneous agents are added. It can significantly outperforms late fusion by 18.3% in AP70 with our feature-level fusion.
>
> | **AP50** | - | - | - | - |
> |--|--|--|--|--|
> | **Methods** | $\mathbf{L}_p^{(64)}$ | $\mathbf{L}_p^{(64)}$ +$\mathbf{C}^{(384)}_E$ | $\mathbf{L}_p^{(64)}$ +$\mathbf{C}^{(384)}_E$ +$\mathbf{L}^{(64)}_S$ | $\mathbf{L}_p^{(64)}$ +$\mathbf{C}^{(384)}_E$ +$\mathbf{L}^{(64)}_S$ +$\mathbf{C}^{(336)}_R$ |
> | F-Cooper | 0.632 | 0.637 | 0.561 | 0.573 |
> | DiscoNet | 0.367 | 0.346 | 0.354 | 0.351 |
> | AttFusion | 0.637 | 0.641 | 0.641 | 0.643 |
> | CoBEVT | 0.613 | 0.641 | 0.653 | 0.646 |
> | HMViT | 0.612 | 0.610 | 0.694 | 0.706 |
> | V2XViT | 0.609 | 0.678 | 0.750 | 0.743 |
> | Late Fusion | 0.632 | 0.666 | 0.733 | 0.725 |
> | HEAL | **0.658** | **0.719** | **0.763** | **0.763** |
> | **AP70** | - | - | - | - |
> | **Methods** | $\mathbf{L}_p^{(64)}$ | $\mathbf{L}_p^{(64)}$ +$\mathbf{C}^{(384)}_E$ | $\mathbf{L}_p^{(64)}$ +$\mathbf{C}^{(384)}_E$ +$\mathbf{L}^{(64)}_S$ | $\mathbf{L}_p^{(64)}$ +$\mathbf{C}^{(384)}_E$ +$\mathbf{L}^{(64)}_S$ +$\mathbf{C}^{(336)}_R$ |
> | F-Cooper | 0.466 | 0.453 | 0.353 | 0.353 |
> | DiscoNet | 0.307 | 0.300 | 0.293 | 0.284 |
> | AttFusion | 0.474 | 0.478 | 0.476 | 0.477 |
> | CoBEVT | 0.412 | 0.420 | 0.398 | 0.372 |
> | HMViT | 0.415 | 0.430 | 0.494 | 0.493 |
> | V2XViT | 0.450 | 0.499 | 0.554 | 0.549 |
> | Late Fusion | 0.473 | 0.492 | 0.569 | 0.558 |
> | HEAL | **0.531** | **0.587** | **0.660** | **0.660** |
>
> - OPV2V (culver city)
>
> Since the OPV2V (culver city) has at most 3 agents in the scene, so we prepare 3 agent types in total.
>
> From the table below, we see that when new heterogeneous agents join the collaboration, HEAL notably outperforms other methods thanks to our effective training strategy and fusion module.
>
> | AP50 | - | - | - |
> |---|--|--|--|
> | **Methods** | $\mathbf{L}_p^{(64)}$ | $\mathbf{L}_p^{(64)}$ +$\mathbf{C}^{(384)}_E$ | $\mathbf{L}_p^{(64)}$ +$\mathbf{C}^{(384)}_E$ +$\mathbf{C}^{(336)}_R$ |
> | F-Cooper | 0.548 | 0.539 | 0.603 |
> | DiscoNet | **0.624** | 0.625 | 0.622 |
> | AttFusion | 0.609 | 0.609 | 0.618 |
> | CoBEVT | 0.606 | 0.622 | 0.647 |
> | HMViT | 0.59 | 0.602 | 0.614 |
> | V2XViT | 0.602 | 0.631 | 0.649 |
> | Late Fusion | 0.569 | 0.589 | 0.616 |
> | HEAL | 0.588 | **0.636** | **0.656** |
> | **AP70** | - | - | - |
> | **Methods** | $\mathbf{L}_p^{(64)}$ | $\mathbf{L}_p^{(64)}$ +$\mathbf{C}^{(384)}_E$ | $\mathbf{L}_p^{(64)}$ +$\mathbf{C}^{(384)}_E$ +$\mathbf{C}^{(336)}_R$ |
> | F-Cooper | 0.46 | 0.432 | 0.438 |
> | DiscoNet | **0.503** | 0.505 | 0.505 |
> | AttFusion | 0.492 | 0.494 | 0.483 |
> | CoBEVT | 0.461 | 0.467 | 0.488 |
> | HMViT | 0.44 | 0.45 | 0.46 |
> | V2XViT | 0.459 | 0.483 | 0.487 |
> | Late Fusion | 0.428 | 0.441 | 0.454 |
> | HEAL | 0.487 | **0.522** | **0.533** |
>
> In conclusion, HEAL presents two main advanatages over existing methods. (i) Comparing with collective training methods, it eliminates intractable multi-agent data collection with new and existing agent types, and significantly reduces the training costs. (ii) Comparing with late fusion, it shows noticeable performance gain with similar training costs.

---

> ### Author Response · Authors · 2023-11-19
> **Response #2**
>
> > **Q1:** In this paper, the authors claim that agents newly joining the system may struggle to align well in the feature space due to data distribution differences. However, is training a unified feature space an effective solution? Given that data discrepancies arising from different sensors inherently result in domain differences, this discrepancy poses a significant challenge in the domain adaptation field. Is the method proposed in this paper suitable for addressing this issue?
>
> Training a unified feature space is an very effective solution when we want the high performance leveraged by feature fusion. Notably, aligning feature from different modalities to a shared space is an emerging practice. Existing single-agent multi-modality detection works[3,4] have achieved substantial success in unifying the BEV features from LiDAR and cameras to facilitate multi-modality perception. X-Align[3] proposes a Cross-Modal Feature Alignment loss to align LiDAR and camera BEV features for more accurate BEV segmentation. BEVDistill[4] uses a dense feature distillation to promote the similarity between the camera feature and the LiDAR feature. These papers all demonstrated that such unified feature space can brought considerable performance enhancements.
>
> In our open heterogeneous setting, this approach particularly shows its superiority. As delineated in Section 3 of our paper, we **cannot predetermine the sensor modality, sensor configuration, and detection model of the new agents**. A unified feature space constitutes a consistent objective for the heterogeneous alliance, simplifying the integration of new agents. They only need to adapt their own features to this unified space. Compare with collective training, it does not require cumbersome data collection and heavy training. Compared with late fusion, it shows higher performance and can be more resistant to noise.
>
>
>
> > **Q2:** The author raises a novel question, thus it would be prudent to utilize more datasets to verify the efficacy of the proposed method. This is because some schemes[7] solely employing distillation can achieve significant improvements in accuracy. In the open-source datasets they used, there are also newly joining agents, similar to the DAIR-V2X dataset used in this paper. It is hoped that the author can supplement with more extensive experiments to substantiate the reliability of the raised question and the effectiveness of the proposed method.
>
> Thanks for your suggestion; we have conducted additional experiments on V2XSet and OPV2V(culver city). See the response to **W2**. Experiment results show that HEAL holds the best performance compared with other intermediate fusion methods and late fusion. Note that in real-world deployment, we have also reduced the cost of data collection because we do not need to collect spatio-temporal synchronous cooperative data for new agents and existing agents in the same scene to perform collective training. It also significantly surpasses late fusion in terms of AP50 & AP70.
>
> Knowledge distillation is similar to other collective training methods, which bring heavy costs associated with data collection and training in real-world scenarios. When a new heterogeneous agent is to be integrated into the scene (acting as a student), it necessitates the presence of an existing agent type (acting as a teacher) to collect temporally synchronized sensor data together.  The open heterogeneous setting has never been considered in terms of accuracy alone. A significant contribution of HEAL is its capacity to prevent the complexities involved in data acquisition and collective training processes. Meanwhile, it can remarkably outperforms late fusion in detection performance.

---

> ### Author Response · Authors · 2023-11-19
> **Response #3**
>
> > **Q3:** Referring to the article on late fusion[5], I believe that late fusion seems to be a more effective solution to the problem posed in this paper. While the process of late fusion indeed has some issues with error accumulation, [5] has adeptly mitigated some of the past problems of late fusion through trajectory prediction. At the same time, employing late fusion can maximally avoid the issue of aligning features extracted by different agents, fundamentally resolving the problem posed in this paper. I hope that the author can conduct a comparative analysis between the methods of these two papers.
>
> Thanks for your insightful comment. Late fusion can disregard the heterogeneity among all agents and achieve collaborative perception through fusion at the output level, as described in the introduction. However, the performance of late fusion is not optimal. That's why researchers are keen on exploring intermediate fusion methodologies[1,2], surpassing late fusion performance.
>
> Even with some techniques to correct the error accumulation, late fusion is still suboptimal against intermediate fusion. It is because (i) the intermediate feature itself is more robust than bounding boxes when facing spatial misalignment (ii) the correction algorithm using bounding boxes can also be applied to feature-level fusion (as CoBEVFlow[5] and CoAlign[6] do).
>
> CoBEVFlow[5] is actually not late fusion but a hybrid collaborative framework that concurrently transmits bounding boxes and intermediate features. Bounding boxes from different timestamps are amalgamated to predict a BEV flow, which is then utilized for **feature warping** to achieve more precise feature fusion results. As illustrated in [5]'s Table 2, the performance of the fusing bounding box is significantly lower than that of intermediate feature fusion:
>
>
> | CoBEVFlow[5]'s Table 2 | - | AP@0.50 / AP@0.70 | AP@0.50 / AP@0.70 |
> |------------------------|--|-------------------|-------------------|
> | **Flow Prediction for Correction** | **Fusion** | **300ms Latency** | **500ms Latency** |
> | $\checkmark$ | Box (late fusion) | 0.764 / 0.611 | 0.571 / 0.399 |
> | $\checkmark$ | Feature (intermediate fusion) | **0.831 / 0.757** | **0.815 / 0.687** |
>
>
>
> CoAlign[6] also transmits bounding boxes with features. They found that the most effective approach is to use bounding boxes to correct noisy poses, but still use feature fusion to obtain higher detection performance. Similar experimental results can be found in CoAlign[6]'s Table 3:
>
> | CoAlign[6]'s Table 3 | - | AP@0.70 | AP@0.70 | AP@0.70 |
> |----------------------|--|---------|---------|---------|
> | **Agent-Object Pose Correction** | **Fusion** | **Noise std (0.2m/0.2 deg)** | **Noise std (0.4m/0.4 deg)** | **Noise std (0.6m/0.6 deg)** |
> | $\checkmark$ | Box (late fusion) | 0.818 | 0.758 | 0.672 |
> | $\checkmark$ | Feature (intermediate fusion) | **0.900** | **0.889** | **0.868** |
>
> Therefore, regarding HEAL, late fusion, and knowledge distillation, we present the following comparative analysis. The performance comparison with late fusion can be found in the paper's Table 3.
>
>
> | Term | HEAL | Late Fusion | Knowledge Distillation |
> |------|------|-------------|------------------------|
> | **No Multi-agent Synchronized Data Collection for New Agent** (low data collection cost) | $\checkmark$ | $\checkmark$ | $\times$ |
> | **No Collective Training** (low training cost) | $\checkmark$ | $\checkmark$ | $\times$ |
> | **High Performance** | $\checkmark$ | $\times$ | $\checkmark$ |
> | **Robustness to Interference** (without correction) | $\checkmark$ | $\times$ | $\checkmark$ |
> | **Robustness to Interference** (with correction) | $\checkmark$$\checkmark$ | $\checkmark$ | $\checkmark$$\checkmark$ |
>
>
> Reference
>
> [1] V2X-ViT: Vehicle-to-Everything Cooperative Perception with Vision Transformer, ECCV 2022
>
> [2] OPV2V: An Open Benchmark Dataset and Fusion Pipeline for Perception with Vehicle-to-Vehicle Communication, ICRA 2023
>
> [3] X-Align: Cross-Modal Cross-View Alignment for Bird's-Eye-View Segmentation, WACV 2023
>
> [4] BEVDistill: Cross-Modal BEV Distillation for Multi-View 3D Object Detection, ICLR 2023
>
> [5] Asynchrony-Robust Collaborative Perception via Bird's Eye View Flow, NeurIPS 2023
>
> [6] Robust Collaborative 3D Object Detection in Presence of Pose Errors, ICRA 2023
>
> [7] MKD-Cooper: Cooperative 3D Object Detection for Autonomous Driving via Multi-teacher Knowledge Distillation. TIV 2023.

---

> > ### Author Response · Authors · 2023-11-23
> > **Response #3 Cont.**
> >
> > We thank the reviewer for providing valuable comments, and we are trying our best to address your concerns further.
> >
> > We have added additional experiments to demonstrate that even with corrections for interference factors such as communication latency or pose errors, late fusion is still suboptimal to intermediate fusion.
> >
> > Here we consider localization noise as an interference factor. CoAlign[1] is the latest collaborative perception framework that resists pose errors. It simultaneously transmits intermediate features and single-agent detected bounding boxes. The bounding boxes will be used to construct an agent-object pose graph to correct the pose for feature fusion. We used the same agent-object pose graph algorithm in the original paper and corrected the pose errors in both late fusion and HEAL. The results are as follows:
> >
> > | w/o Correction                                   |           |           |           |           |           |           |           |
> > | ------------------------------------------------ | --------- | --------- | --------- | --------- | --------- | --------- | --------- |
> > | **Pose Errors Std. (m/deg)**                     | 0/0       | 0.2/0.2   | 0.4/0.4   | 0.6/0.6   | 0.8/0.8   | 1.0/1.0   | 1.2/1.2   |
> > | **AP50**                                         |           |           |           |           |           |           |           |
> > | Late Fusion                                      | 0.834     | 0.735     | 0.491     | 0.392     | 0.354     | 0.335     | 0.335     |
> > | HEAL                                             | **0.894** | **0.869** | **0.802** | **0.747** | **0.716** | **0.692** | **0.683** |
> > | **AP70**                                         |           |           |           |           |           |           |           |
> > | Late Fusion                                      | 0.685     | 0.398     | 0.256     | 0.232     | 0.230     | 0.232     | 0.236     |
> > | HEAL                                             | **0.813** | **0.712** | **0.586** | **0.532** | **0.523** | **0.524** | **0.528** |
> > |                                                  |           |           |           |           |           |           |           |
> > | **w/ Correction [1]:**                           |           |           |           |           |           |           |           |
> > | **Pose Errors Std. (m/deg)**                     | 0/0       | 0.2/0.2   | 0.4/0.4   | 0.6/0.6   | 0.8/0.8   | 1.0/1.0   | 1.2/1.2   |
> > | **AP50**                                         |           |           |           |           |           |           |           |
> > | Late Fusion + Agent-Object Pose Graph Correction | 0.834     | 0.806     | 0.749     | 0.670     | 0.581     | 0.507     | 0.461     |
> > | HEAL + Agent-Object Pose Graph Correction        | **0.894** | **0.884** | **0.865** | **0.840** | **0.800** | **0.774** | **0.739** |
> > | **AP70**                                         |           |           |           |           |           |           |           |
> > | Late Fusion + Agent-Object Pose Graph Correction | 0.685     | 0.604     | 0.571     | 0.508     | 0.441     | 0.381     | 0.346     |
> > | HEAL + Agent-Object Pose Graph Correction        | **0.813** | **0.781** | **0.751** | **0.724** | **0.679** | **0.644** | **0.616** |
> >
> > We see that (i) without agent-object pose graph correction, HEAL significantly exceeds late fusion under various noise conditions (especially large noise), which shows the unreliability of only fusing bounding boxes. (ii) With agent-object pose graph correction, both late fusion and HEAL significantly improve detection accuracy, but HEAL remarkably outperforms late fusion and is more robust across various noise levels. Since the correction is compensated at the feature level, using feature fusion can more effectively resist cumulative errors.
> > In conclusion, HEAL's design with feature-level collaboration is the key to achieving high performance and robust collaborative perception. Further, we maintain a training cost similar to late fusion, which proves the effectiveness of our new agent integration with high performance.
> >
> > [1] Robust Collaborative 3D Object Detection in Presence of Pose Errors. ICRA 2023

---

> ### Author Response · Authors · 2023-11-22
>
> We extend our gratitude for your invaluable critique. In responding, we  have strived to comprehensively address each of your concerns and  queries:
>
> - In our discourse, we have elucidated the rationale for opting for intermediate fusion, as demonstrated in numerous studies [1,2,3,4,5,6,7,8,9,10] on collaborative perception to yield superior detection performance. Additionally, the training parameters, FLOPs, and other metrics of  HEAL's new agent types are all comparable to those of late fusion, underscoring the efficiency of our novel framework design. Thus, HEAL can integrate new heterogenous agents as effectively as late fusion, with significantly better performance.
> - Utilizing bounding box information to mitigate cumulative errors is indeed viable. However, current methodologies[8,9] tend to apply such box-based correction methods at the feature level instead of merely at the box level to get more robust performance. Therefore, obtaining robust high performance still depends on feature fusion.
> - In accordance with your request, we have augmented our manuscript with comprehensive experiments on two additional datasets. These experiments further validate HEAL's performance superiority, employing intermediate fusion over late fusion.
>
> May we request you to kindly review our revised response and confirm whether it adequately addresses your inquiries? We remain open to further dialogue and are prepared to provide additional explanations for any new questions you might raise.
>
>
>
> References:
>
> [1] Opv2v: An open benchmark dataset and fusion pipeline for perception with vehicle-to-vehicle communication, ICRA 2022.
>
> [2] V2X-ViT: Vehicle-to-everything cooperative perception with vision transformer. ECCV 2022.
>
> [3] V2v4real: A real-world large-scale dataset for vehicle-to-vehicle cooperative perception. CVPR 2023
>
> [4] V2xp-asg: Generating adversarial scenes for vehicle-to-everything perception. ICRA 2023
>
> [5] Collaboration Helps Camera Overtake LiDAR in 3D Detection. CVPR 2023
>
> [6] HM-ViT: Hetero-modal Vehicle-to-Vehicle Cooperative perception with vision transformer, ICCV 2023
>
> [7] CoBEVT: Cooperative bird's eye view semantic segmentation with sparse transformers. CoRL 2022.
>
> [8] Robust Collaborative 3D Object Detection in Presence of Pose Errors. ICRA 2023
>
> [9] Asynchrony-Robust Collaborative Perception via Bird’s Eye View Flow. NeurIPS 2023.
>
> [10] Where2comm: Communication-Efficient Collaborative Perception via Spatial Confidence Maps. NeruIPS 2022

---

### Author Response · Authors · 2023-11-19
**HEAL's real-world significance**

We hope that reviewers recognize the real-world significance of this paper. Every aspect of our HEAL is designed to facilitate open heterogeneous cooperative perception in the real world, aiming to reduce the cost of deploying the V2V systems:

**(i) Data Collection Costs:** Previous works with collective training require collecting temporally and spatially synchronized multi-agent sensor data for new and existing agents. This issue is not evident in the OPV2V-H and DAIR-V2X datasets but represents an unavoidable real-world challenge. HEAL allows autonomous driving companies to train with data from single agents, eliminating the constant need to collect synchronized data with new and existing heterogeneous agent types.

**(ii) Training Costs and Performance:** We have conducted a rigorous analysis comparing our model's parameters, FLOPs, training throughput, and memory usage with all baseline methods. For previous methods, the training costs escalate considerably with the increase in agent types. Our experiments demonstrate that HEAL maintains optimal performance while minimizing these training costs.

**(iii) Model Detail Privacy Concerns:** All previous works overlook this critical point. When deploying cooperative perception in real-world scenarios, would new heterogeneous vehicles be willing to share their model details for collective training? HEAL allows autonomous driving companies to train locally with their sensor data, thereby maximizing privacy protection and addressing potential concerns about deploying V2V technology.

These considerations underscore HEAL's practicality and potential to impact the future real-world deployment of heterogeneous collaborative perception significantly. To sum up, HEAL offers solution-oriented and practical approaches to the industry.

---

### Meta-Review · Area_Chair_FdQs · 2023-12-17

**Metareview:**

The proposed multi-agent collaborative perception is a new and interesting task. The proposed  HEAL framework is capable of accommodating the features acquired by the agents newly joining the system. Most reviewers give positive comments. Reviewer bqtt  and authors conduct extensive discussion. Afer looking into discussion details, it can be found that authors conducted comprehensive and extensive experiments that demonstrate the consistent superior detection performance of HEAL and provided a thorough explanation for our choice of intermediate fusion and the architectural design differences between HEAL and previous methods. Considering the real-world significance of HEAL, the AC decided to accept it.

**Justification For Why Not Higher Score:**

More comparsions with broader multi-agent algorithems can be added.

**Justification For Why Not Lower Score:**

The task is inspiring.

---

### Decision · Program_Chairs · 2024-01-16

Accept (poster)